# Tiny Brains, Giant Impact: Uncovering the Keystone Neurons of LLM with Just a Few Prompts

Xiangtian Ji [* 1]  Yuxin Chen [* 1]  Zhengzhou Cai [2]  Xiang Wang [3]  An Zhang [3]  Tat-Seng Chua [1]

## Abstract

Large language models (LLMs) display strong comprehensive abilities, yet the internal mechanisms that support these behaviors remain insufficiently understood. In this work, we show that across a wide range of open-weight Transformers, a subset of neurons remains consistently highly activated during inference across tasks of multiple capability dimensions. By probing along the cross-task activation strength, an extremely sparse subset is isolated, whose removal causes a collapse in model behavior, which we term keystone neurons. Our analysis reveals that keystone neurons are a stable and intrinsic neuron subset of the model that is largely established during pretraining. The parameters associated with these neurons are tightly calibrated during the training process, and their precise values are critical for the capabilities of the model. Building on these insights, we propose a supervised fine-tuning approach that updates only keystone neurons, achieving task gains comparable to or even better than full-parameter fine-tuning while better preserving performance in other capability dimensions, despite modifying a much smaller number of parameters.

## 1. Introduction

Large language models (LLMs) have achieved impressive performance across a broad spectrum of real-world tasks, often reaching or even surpassing human-level capabilities (Brown et al., 2020; Chowdhery et al., 2022; OpenAI, 2024; Bubeck et al., 2023; Lewkowycz et al., 2022; Li et al., 2022; Zhao et al., 2025; DeepSeek-AI, 2025; Team, 2026; Zhang et al., 2024; 2026; Liu et al., 2025). Despite this

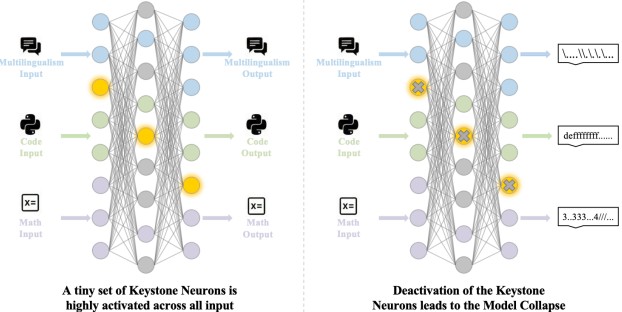

*Figure 1.* Conceptual illustration of keystone neurons. These neurons are consistently engaged across diverse prompts, and their deactivation leads to global capability collapse.

rapid progress on external capabilities, the understanding of the internal mechanisms that give rise to these behaviors remains limited. A growing body of work shows that LLMs do not rely on all parameters in the same way, but instead develop stable patterns of specialization across their internal structure (Qiu et al., 2024; Fedus et al., 2022; Zoph et al., 2022; Bricken et al., 2023; Pochinkov, 2023).

Subsequent work shows that several studies identify neurons whose activations are systematically associated with particular tasks—such as multilingual processing (Tang et al., 2024; Chen et al., 2025; Zhao et al., 2024), code generation (Miller et al., 2025), and mathematical reasoning (Yu and Ananiadou, 2024)—and show that interventions on these units predominantly affect the corresponding abilities, with comparatively limited impact on unrelated behaviors (Song et al., 2024; Wang et al., 2025a).

These task-specific insights compel a more fundamental structural inquiry: *beyond task-specific neurons, does a model rely on a tiny set of neurons that is repeatedly engaged across diverse prompts and is necessary for broad capability execution?* In Figure 2, we observe a group of neurons that remains consistently highly activated across diverse capability dimensions on Qwen2.5-7B-Instruct. By ranking neurons by their cross-task activation and gradually expanding the subset, we encounter a very sparse frontier whose removal already causes severe degradation of the model's abilities, including basic language modeling. Remarkably, the same pattern also appears across multiple

---

*Equal contribution [1]National University of Singapore [2]Beijing University of Posts and Telecommunications [3]University of Science and Technology of China. Correspondence to: An Zhang <anzhang@ustc.edu.cn>.

*Proceedings of the 43rd International Conference on Machine Learning*, Seoul, South Korea. PMLR 306, 2026. Copyright 2026 by the author(s).

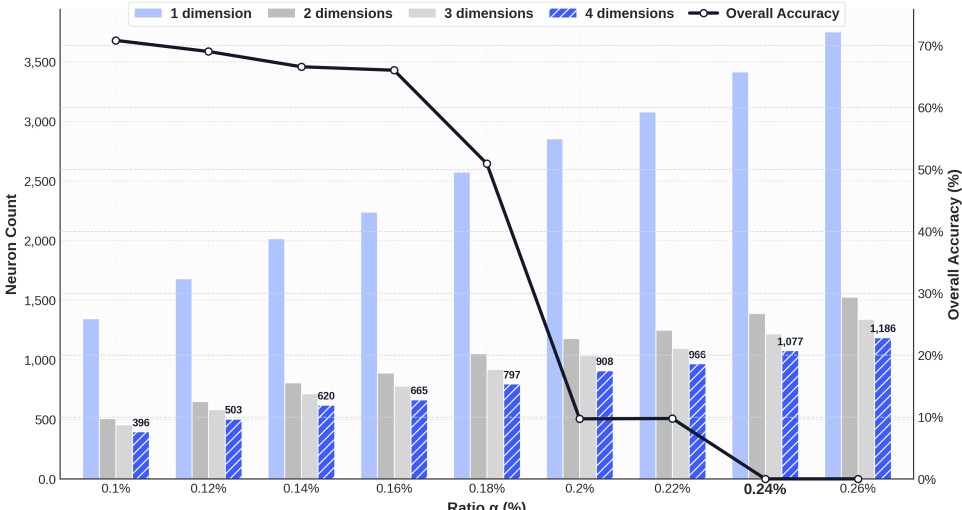

*Figure 2.* Neuron intersections and accuracy for Qwen2.5-7B-Instruct when selecting the top-$\alpha$ fraction of neurons ranked by activation across four capability dimensions. Bars show, for each $\alpha$, how many neurons appear in the top-$\alpha$ sets for 1, 2, 3, or all 4 dimensions, while the black curve (right axis) reports the model comprehensive capability after masking the neurons in the 4-dimension intersection.

model families, suggesting that a sparse, cross-task–active neuron subset exists within these models as a foundation of modern LLM behavior. Building on these observations, our focus shifts from task-specific units to neurons with broad functional importance, as illustrated in Figure 1, which we term *keystone neurons*.

Across a wide range of open-weight LLMs, we identify a tiny keystone-neuron set using only *four* prompts that span different capability dimensions. This set is extremely sparse (typically $< 0.2\%$ of all neurons) yet highly stable under prompt resampling. For example, in Llama 3.1-8B, we identify only 30 keystone neurons out of 1,245,184 neurons in total. Repeating the same four-dimension identification with alternative prompt sets yields consistently high overlaps, suggesting that keystone neurons reflect an intrinsic model property rather than prompt-specific artifacts.

We then assess the functional importance of keystone neurons through two classes of interventions: inference-time perturbations and targeted fine-tuning. For inference-time perturbations, we modify only the outputs of keystone neurons by multiplicative rescaling while leaving the rest of the network unchanged. This subset exhibits high sensitivity to parameter scaling, indicating that the precise calibration of their outputs is essential for maintaining overall model performance. For targeted fine-tuning, updating only weights associated with keystone neurons yields larger gains than full fine-tuning updates in both mathematical reasoning and safety safeguarding task, while better preserving other capabilities. Overall, these results indicate that, beyond neurons tied to specific task, LLMs include a compact set of neurons whose influence spans multiple capabilities and on which broad model behavior disproportionately depends.

## 2. Keystone Neurons Identification

This section formalizes the study setting and the procedure for identifying *keystone neurons* in LLMs. Intuitively, the goal is to identify neurons that remain strongly activated across diverse tasks and whose removal leads to substantial degradation of overall model functionality.

### 2.1. Preliminaries

We consider a standard decoder-only Transformer with $L$ stacked layers, each containing a self-attention module and a feed-forward (FFN) module. Every learnable linear projection—including FFN up- and down-projections and the attention-projection matrices used to compute queries, keys, values, and output representations in the self-attention mechanism—is treated as a collection of neurons. Concretely, each row (or column) of these matrices is referred to as a neuron, representing a one-dimensional feature channel that linearly projects the input representation (Frankle and Carbin, 2019; Wang et al., 2025b).

We index all neurons by a single index $i \in \{1, \ldots, N\}$, where $N$ is the total number of neurons across all layers and modules. Each neuron $i$ belongs a layer–module block (e.g., a specific FFN-up or attention-$Q$ matrix). Given the vector input to the neuron's linear projection at decoding step $t$, the scalar activation of neuron $i$ is denoted by $a_i(t)$. This quantity is the output of the corresponding linear channel applied to the hidden state (before any subsequent nonlinearity or normalization in the block).

*Table 1.* Representative neuron-masking results. Full results are provided in Appendix C.1

| Model | Setting | Accuracy ↑ | | | | Perplexity ↓ | |
|---|---|---|---|---|---|---|---|
| | | MMLU | Math500 | MGSM | EvalPlus | C4 | W2 |
| **Qwen3-8B** | Base | 0.821 | 0.942 | 0.842 | 0.669 | 15.74 | 15.69 |
| | Keystone-off | **0** | **0** | **0** | **0** | 19.53 | 23.74 |
| | Random-off | 0.824 | 0.638 | 0.856 | 0.651 | 17.15 | 17.43 |
| **Qwen3-0.6B** | Base | 0.530 | 0.720 | 0.684 | 0.423 | 29.97 | 33.73 |
| | Keystone-off | **0** | **0** | **0** | **0** | 340 | 566 |
| | Random-off | 0.518 | 0.540 | 0.616 | 0.431 | 32.28 | 39.37 |
| **Llama-3.1-8B-Inst.** | Base | 0.713 | 0.496 | 0.792 | 0.532 | 11.78 | 13.61 |
| | Keystone-off | **0** | **0** | **0** | **0** | 522 | 879 |
| | Random-off | 0.706 | 0.486 | 0.770 | 0.545 | 11.83 | 17.76 |
| **Llama-3.2-1B-Inst.** | Base | 0.410 | 0.236 | 0.300 | 0.246 | 11.83 | 17.77 |
| | Keystone-off | **0** | **0** | **0** | **0** | 779 | 807 |
| | Random-off | 0.401 | 0.204 | 0.262 | 0.235 | 21.01 | 24.52 |
| **Gemma-3-1B-IT** | Base | 0.387 | 0.438 | 0.468 | 0.386 | 297 | 299 |
| | Keystone-off | **0** | **0** | **0** | **0** | 50938 | 130328 |
| | Random-off | 0.373 | 0.440 | 0.462 | 0.386 | 255.31 | 114.76 |
| **Qwen3-30B-A3B-Inst.** | Base | 0.865 | 0.972 | 0.930 | 0.698 | 14.77 | 12.10 |
| | Keystone-off | **0** | **0** | **0** | **0** | 206.91 | 152.10 |
| | Random-off | 0.842 | 0.826 | 0.854 | 0.669 | 14.46 | 12.84 |
| **Mixtral-8×7B-Inst.** | Base | 0.645 | 0.284 | 0.528 | 0.476 | 32.12 | 11.14 |
| | Keystone-off | **0** | **0** | **0** | **0** | 580 | 250 |
| | Random-off | 0.609 | 0.248 | 0.490 | 0.402 | 47.35 | 12.90 |

## 2.2. Detection of keystone neurons

**Stage 1: Multi-prompt activation analysis.** We construct a small probe set of prompts $\mathcal{P} = \{p_1, \ldots, p_K\}$, each chosen to elicit a different model capability. For each prompt $p \in \mathcal{P}$, the model generates a full response, and we record the activation $a_i(p, t)$ of every neuron $i$ at each decoding step $t$. To summarize how strongly neuron $i$ participates in the response to prompt $p$, we define $\bar{a}_i(p)$ as the average of $|a_i(p, t)|$ over all generated tokens $t$. This quantity provides a simple measure of neuron $i$'s contribution to the model's processing of prompt $p$. We then group neurons by their layer and module block. Within each block and for each prompt, neurons are sorted by $\bar{a}_i(p)$, and we keep only the top $\rho$ fraction in that block for that prompt. Repeating this over all prompts produces multiple top-activation sets. Finally, we take the intersection of these sets across all prompts. The surviving neurons form the candidate pool $\mathcal{S}_{\text{cand}}$, consisting of neurons that remain among the most strongly activated ones across diverse capability probes.

**Stage 2: $\alpha$-controlled neuron masking.** Stage 1 produces, for each model, a ranked list of neurons based on their multi-prompt activation statistics. We now use this ranking to progressively expand along the high-activation direction and test how these consistently activated neurons affect model behavior through controlled masking experiments. We introduce a scalar hyperparameter $\alpha \in (0, 1)$ that specifies the fraction of top-ranked neurons to intervene on. For a given model and a chosen value of $\alpha$, we select the top $\alpha$ fraction of neurons from its ranked list and denote this subset by $\mathcal{S}_{\text{top}}(\alpha^\star)$. During evaluation, neurons in $\mathcal{S}_{\text{top}}(\alpha^\star)$ are deactivated by zeroing out their output contributions in the forward pass, while all other parameters remain unchanged. For each model, we sweep over a small

set of $\alpha$ values (corresponding to different, but always very small, fractions of the total neuron count) and measure performance on a shared suite of evaluation benchmarks that cover different capabilities.

## 3. Keystone Neurons Analysis

### 3.1. Experiment setup

**Models.** We study keystone neurons across several open-weight transformer families. Specifically, we evaluate the Qwen3 series, including Qwen3-0.6B, Qwen3-8B, and the 30B MoE variant Qwen3-30B-A3B and its Qwen3-30B-A3B-Instruct (Yang et al., 2025); the Qwen2.5 series, including Qwen2.5-0.5B, Qwen2.5-0.5B-Instruct, Qwen2.5-7B, and Qwen2.5-7B-Instruct (Qwen et al., 2025); the Gemma3 series, including Gemma-3-1B base and Gemma-3-1B-Instruct models (Gemma Team, 2025); the Llama3 series, including Llama-3.2-1B, Llama-3.2-1B-Instruct, Llama-3.1-8B, and Llama-3.1-8B-Instruct (Llama3 Team, 2024); a set of reasoning-distilled DeepSeek-R1 models built on these backbones, including DeepSeek-R1-Distill-Qwen-1.5B, DeepSeek-R1-Distill-Qwen-7B, and DeepSeek-R1-Distill-Llama-8B (DeepSeek-AI, 2025); and the Mixtral series, including Mixtral-8×7B-v0.1 and Mixtral-8×7B-v0.1-Instruct (Jiang et al., 2024).

**Benchmarks.** We use four core benchmarks in two roles: MMLU for general tasks, MATH500 for mathematics, EvalPlus for coding-style program synthesis, and MGSM for multilingual tasks (Hendrycks et al., 2021a;b; Liu et al., 2023; Shi et al., 2022). As tasks along general tasks, mathematical reasoning, code generation, and multilingual tasks are commonly emphasized in contemporary LLM evaluation, we use these four capabilities as the main evaluation axes in our experiments (Yang et al., 2025; Qwen et al.,

2025). Representative prompts from these benchmarks serve as the four capability dimensions in our keystone-neuron detection procedure, and we also evaluate model accuracy on the same benchmarks when measuring the effect of neuron masking. We additionally report perplexity on C4 and WikiText-2 to track language modeling quality (Raffel et al., 2023; Dodge et al., 2021; Merity et al., 2016).

## 3.2. Keystone neurons: a compact capability backbone

**Keystone neurons form a tiny yet structurally critical subset of parameters.** For each model, the detection procedure in Section 2 yields a keystone set that occupies only a small fraction of all neurons. Table 2 summarizes these sizes together with their prompt-level stability. Across model families, the absolute number of keystone neurons ranges from a few dozen in small dense models up to a few thousand in larger or MoE models, yet their share of all neurons typically stays well below 0.2%. Despite their small size, zeroing keystone neurons causes the model to *lose all capabilities*; for example, in LLaMA-3.1-8B-Instruct, deactivating merely 0.0072% of neurons is sufficient to trigger a complete capability collapse. In contrast, deactivating an equal number of randomly selected neurons typically preserves accuracy close to the base model and induces only moderate perplexity changes (Table 1). This indicates that keystone neurons are not an arbitrary high-activation subset, but a structurally indispensable backbone that supports model capability. Additional controls in Appendix C.3 show that this collapse is not reproduced by high-norm neurons, module-aware random neurons, or substantially larger random ablation budgets.

**Keystone neurons are intrinsic to the model and remain stable across prompt choices.** To test whether keystone neurons depend sensitively on the particular prompts used for detection, the Stage 1 activation analysis is repeated with five disjoint prompt groups for each model. Each group covers the same four capability dimensions (general tasks, mathematical reasoning, code generation, multilingual tasks) but uses non-overlapping instructions. For every model, the pairwise intersection-over-union (IoU) between the five detected keystone sets is computed, and the average over all prompt-group pairs is reported in Table 2. Most models fall in the 80%–95% range, and even the lowest score remains above 73%. In other words, replacing the detection prompts with entirely different surface content within the same capability dimensions consistently recovers essentially the same keystone subset. This prompt-agnostic stability holds for dense, instruction-tuned, reasoning-distilled, and MoE architectures alike, suggesting that keystone neurons are largely determined by the internal organization of the model, while prompts merely act as probes that reveal an already formed backbone.

*Table 2.* Keystone neuron counts and prompt-agnostic stability across models.

| Series | Model | Neurons | Share (%) | IoU Across 5 Groups (%) |
|---|---|---|---|---|
| **Qwen3-dense** | 8B | 126 | 0.0101 | 92.47 |
| | 0.6B | 387 | 0.2383 | 86.47 |
| **Gemma3** | 1B-IT | 96 | 0.0221 | 88.54 |
| | 1B-PT | 218 | 0.0502 | 86.24 |
| **Llama3** | 3.2-1B-Instruct | 42 | 0.0034 | 85.71 |
| | 3.2-1B | 22 | 0.0032 | 95.45 |
| | 3.1-8B-Instruct | 90 | 0.0072 | 97.41 |
| | 3.1-8B | 30 | 0.0024 | 86.67 |
| **Qwen2.5** | 0.5B-Instruct | 69 | 0.0244 | 93.18 |
| | 0.5B | 57 | 0.0202 | 87.04 |
| | 7B-Instruct | 1077 | 0.0835 | 94.13 |
| | 7B | 966 | 0.0749 | 90.60 |
| **DeepSeek-R1-distill** | Llama-8B | 126 | 0.0101 | 91.80 |
| | Qwen-1.5B | 563 | 0.1060 | 81.17 |
| | Qwen-7B | 2090 | 0.1620 | 92.47 |
| **MoE** | Qwen3-30B-A3B-Instruct | 11483 | 0.1189 | 78.69 |
| | Qwen3-30B-A3B | 7001 | 0.0725 | 75.79 |
| | Mixtral-8x7B-v0.1-Instruct | 6400 | 0.0835 | 84.50 |
| | Mixtral-8x7B-v0.1 | 151 | 0.0020 | 73.77 |

## 3.3. Keystone neurons are tightly calibrated

**Keystone neurons exhibit substantially higher sensitivity to multiplicative scaling than random neurons.** We further examine the effect of continuous scaling interventions on Qwen2.5-7B-Instruct and Qwen2.5-0.5B-Instruct. Specifically, during the forward pass, we multiply the outputs of all keystone neurons by a scalar factor $r$, leaving all other neurons unchanged. As a control, we apply the same operation to a randomly selected neuron set of matched size. For each value of $r$, we evaluate accuracy on four benchmarks (MMLU, Math500, MGSM, EvalPlus) and aggregate them with a fixed-weight average into a single comprehensive capability score. Figure 3 reports the comprehensive score as a function of the scaling factor.

Across both keystone and random neuron sets, small perturbations around $r = 1$ can yield minor performance fluctuations within a narrow neighborhood. Beyond this regime, the responses diverge sharply. Scaling keystone neurons produces a rapid and pronounced degradation in the comprehensive capability score: both amplification ($r > 1$) and attenuation ($r < 1$) lead to steep losses under comparatively modest deviations from unity. In contrast, scaling random neurons results in substantially smoother trajectories, with performance remaining comparatively stable over a wider range of $r$ and deteriorating mainly under extreme scaling. Overall, these results indicate that keystone neurons occupy a disproportionately influential operating regime: their outputs are tightly calibrated during the training process, as global multiplicative rescaling disrupts the balance of signals they carry to downstream computation far more severely than for random neurons.

**Keystone neurons require precise inference-time activations rather than representative constant values.** To test

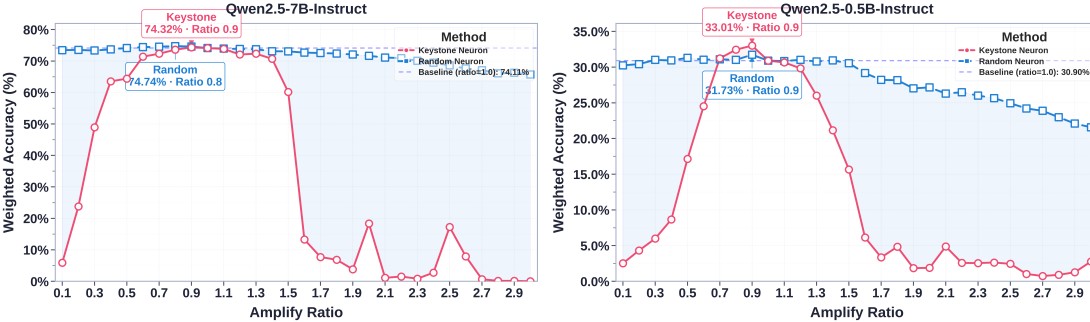

*Figure 3.* Performance under multiplicative rescaling of keystone vs. random neurons in Qwen2.5-7B-Instruct and Qwen2.5-0.5B-Instruct, measured by a fixed-weight aggregate score over MMLU, Math500, MGSM, and EvalPlus.

*Table 3.* Activation patching results on Llama-3.1-8B-Instruct.

| Setting | MMLU | Math500 | MGSM | EvalPlus |
|---|---|---|---|---|
| Native | 0.713 | 0.496 | 0.792 | 0.532 |
| Zero | 0 | 0 | 0 | 0 |
| Response mean | 0 | 0 | 0 | 0 |
| First token | 0 | 0 | 0 | 0 |

whether keystone neurons merely behave as large learned constants, we perform activation patching on Llama-3.1-8B-Instruct. During generation, we replace the native keystone-neuron activations with representative nonzero values computed from the detection prompts: either the response-level mean activation or the activation at the first generated token. As shown in Table 3, both interventions cause complete capability collapse, matching the effect of direct deactivation. Thus, preserving a realistic static value is insufficient; model behavior depends on the precise token-wise activations carried by keystone neurons during inference. Full per-source patching results are provided in Appendix C.2.

### 3.4. Keystone neurons as stable high-activation channels

**Keystone neurons concentrate on a narrow high-activation, low-variability frontier within each module.** Figure 4 indicates that, across all attention and FFN submodules of LLaMA-3.1-8B-Instruct, keystone neurons lie in the tail region characterized by relatively large mean activation and a small coefficient of variation, whereas the vast majority of neurons populate a dense cloud near the origin or reside in a high-variance regime. The reported statistics are computed after within-layer normalization and are aggregated over tokens drawn from four heterogeneous prompts, rendering the observed separation unlikely to be attributable to any single task or a small subset of unusually scaled layers. Taken together, these patterns suggest that keystone neurons constitute a small set of channels that transmit strong yet stable signals across prompts and tokens, forming high-throughput, low-noise pathways within both the residual and attention streams.

### 3.5. The origin of keystone neurons

**Across most model families, keystone neurons are largely established during pre-training, while post-training primarily thickens the existing backbone.** In Figure 5, the Qwen2.5 series exhibits a mild U-shaped pattern with slightly more keystone neurons near the input and output layers and fewer in the middle, whereas the Llama3 series, the Qwen3-MoE series, and the Gemma3 series show a more uniform distribution across depth with only modest variation. Within each family, instruction tuning does not materially alter this profile: base and instruct curves are almost indistinguishable, with aligned peaks/troughs and preserved overall shape. This depth-wise invariance, together with the high base–instruct overlaps observed for Qwen2.5, Qwen3-MoE, and Llama3, supports a picture in which keystone neurons are primarily formed during pre-training, and post-training mainly thickens the same backbone rather than constructing a new set from scratch.

The Gemma3 series follows a different pattern at the level of neuron identity: its base and instruct models share a similar smooth depth profile, but the overlap between their keystone-neuron sets is substantially lower, indicating that many keystone neurons are replaced during alignment.

## 4. Targeted Fine-tuning on Keystone Neurons

Motivated by the preceding analyses, we view keystone neurons as a small set of units that exert disproportionate influence on model behavior and are consistently engaged during decoding across diverse tasks. This leads to the hypothesis that restricting supervised updates to parameters associated with these neurons, instead of updating all model parameters, can yield task-specific improvements while better preserving other capabilities. To test this hypothesis, we compare standard full-parameter supervised fine-tuning with a keystone-only variant that confines updates to the keystone-neuron set identified in Section 2.

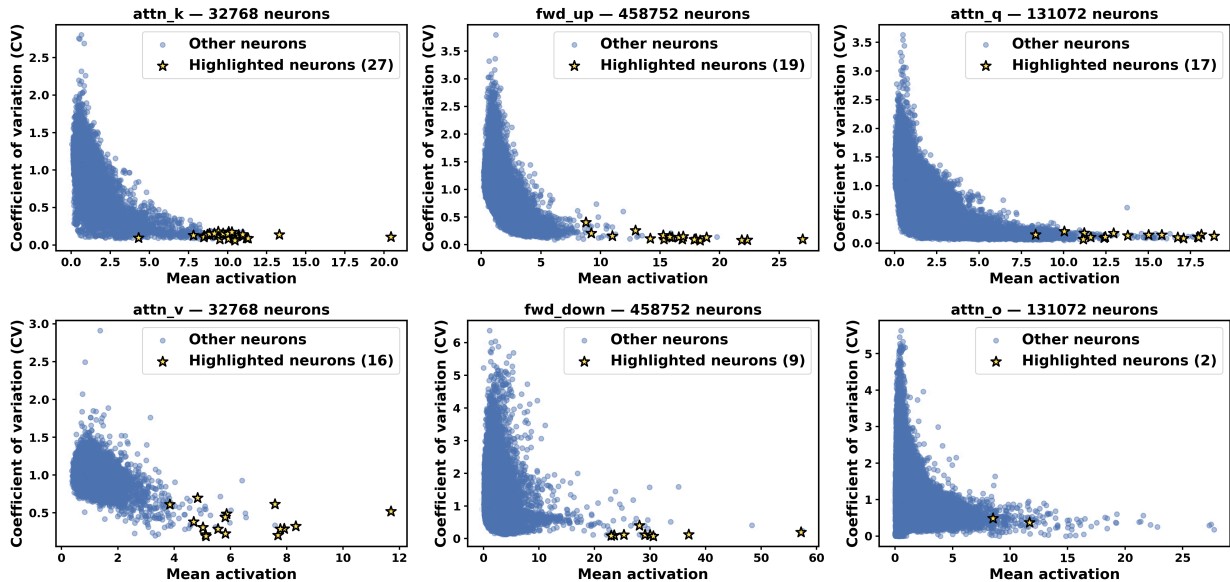

*Figure 4.* Mean–CV activation patterns of keystone neurons in different submodules of Llama-3.1-8B-Instruct. Each panel plots all neurons in one module (self-attention $Q/K/V/O$ projections and FFN up/down projections) as blue dots in the plane of mean activation (x-axis) versus coefficient of variation (CV, y-axis), computed over all generated tokens for the four detection prompts (MMLU, MATH500, EvalPlus, MGSM). Neurons selected into the keystone set $\mathcal{S}_{\text{keystone}}$ are overlaid as yellow stars.

### 4.1. Fine-tuning setup

**Training Data.** Our fine-tuning study primarily targets mathematical reasoning, and additionally includes a safeguarding task to probe behavior on a different downstream objective. For mathematics, models are fine-tuned on 10k instruction–response pairs sampled from OpenMath Instruct 2 (Toshniwal et al., 2024), using exactly the same data and sampling order across all settings. For safeguarding, we use a 10k subset of the WildGuard training corpus, containing prompts, model responses, and human safety judgements that supervise a safeguard model (Han et al., 2024). We fine-tune and evaluate each task separately.

**Models.** The fine-tuning study considers two instruction-tuned Llama models that differ in scale and initial math capability: Llama-3.1-8B-Instruct and Llama-3.2-1B-Instruct (Llama3 Team, 2024).

**Fine-tuning strategies.** For each model, two fine-tuning strategies are evaluated against the original instruction-tuned checkpoint (Base): (1) **Full FT**: standard full-parameter supervised fine-tuning on the math data; (2) **Keystone-only**: fine-tuning where only parameters associated with the keystone neuron set $\mathcal{S}_{\text{keystone}}$ are updated, while all other parameters are frozen. Apart from the set of trainable parameters, Full FT and Keystone-only share identical optimization hyperparameters; full details are provided in Appendix A.

**Evaluation Benchmarks.** Math performance is evaluated on four benchmarks that cover a range of difficulty and style:

MATH500 as a medium-scale competition-style benchmark derived from MATH (Hendrycks et al., 2021b), GSM8K as a grade-school math reasoning benchmark (Cobbe et al., 2021), OlympiadBench and AIME-2024 for harder problems (He et al., 2024; HuggingFaceH4, 2024).

To track changes in broader capabilities, models are also evaluated on three non-math benchmarks: MMLU for general knowledge (Hendrycks et al., 2021a), MGSM for multilingual tasks (Shi et al., 2022), and EvalPlus for coding-style program synthesis (Liu et al., 2023). For each model and setting (Base, Full FT, Keystone-only), accuracy (%) is reported on all seven benchmarks.

### 4.2. Math fine-tuning results

**Keystone-only tuning consistently outperforms full-parameter fine-tuning on mathematical reasoning.** According to results shown in Table 4, across both instruction-tuned Llama models, restricting updates to keystone-associated weights yields consistently stronger math improvements than full-parameter fine-tuning, despite using identical data and optimization budgets. Full-parameter updates often introduce notable regressions—particularly for the 8B model—whereas Keystone-only not only avoids these drops but also restores or surpasses the Base performance across all math evaluations. For the 1B model, where full tuning already brings moderate gains, Keystone-only further enhances performance or matches the best results.

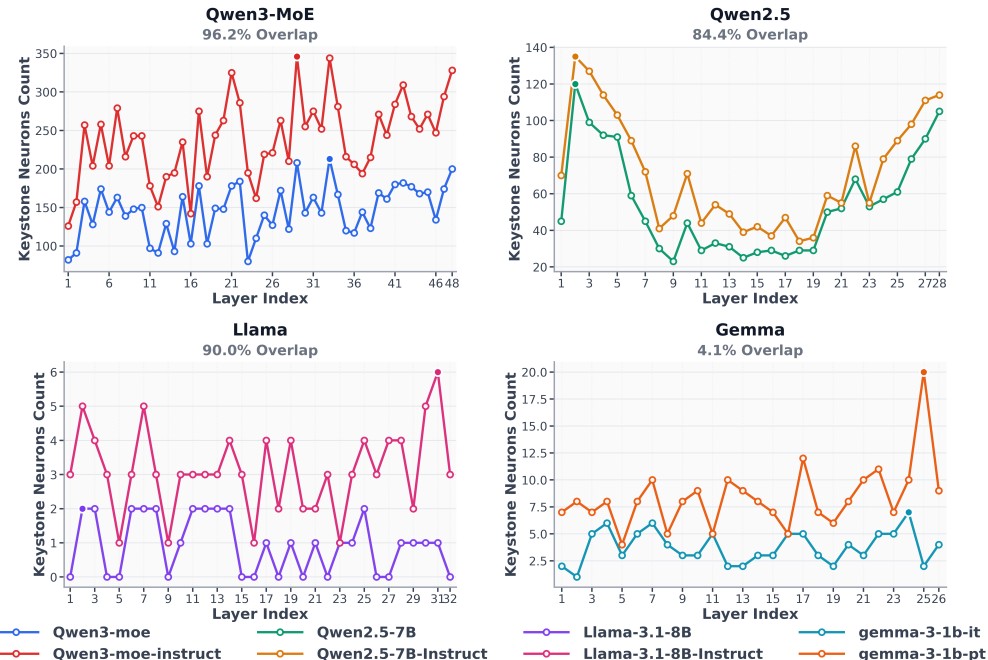

*Figure 5.* Layer-wise counts of keystone neurons for base vs. instruction-tuned models across Qwen2.5, Qwen3-MoE, Llama3, and Gemma3 families; panel titles report base–instruct keystone overlap (%).

*Table 4.* Accuracy (%) on math and non-math benchmarks after 10k math SFT. **Base**: original checkpoint. **Full FT**: full-parameter supervised fine-tuning. **Keystone-only**: SFT updating only $\mathcal{S}_{\text{keystone}}$.

| Model | Setting | Math Benchmarks (↑) | | | | Non-Math Benchmarks (↑) | | |
|---|---|---|---|---|---|---|---|---|
| | | MATH500 | GSM8K | Olympiad | AIME-24 | MMLU | MGSM | EvalPlus |
| **Llama-3.1-8B-Instruct** | Base | 49.60 | 83.40 | 16.59 | 3.33 | **70.9** | **77.2** | 0.532 |
| | Full FT | 31.40 | 70.05 | 8.15 | 0.00 | 62.5 | 60.8 | 0.322 |
| | Keystone-only | **51.20** | **84.31** | **16.74** | **6.67** | 69.8 | 76.6 | **0.550** |
| **Llama-3.2-1B-Instruct** | Base | 23.60 | 29.26 | 3.85 | 0.00 | 41.1 | 32.2 | 0.246 |
| | Full FT | 25.00 | 45.03 | 5.33 | 0.00 | 41.4 | 39.4 | 0.331 |
| | Keystone-only | **29.80** | **50.27** | **6.96** | 0.00 | **44.8** | **39.6** | **0.339** |

**Keystone-only tuning better preserves general capabilities.** On non-math benchmarks, Keystone-only produces far milder distributional shifts than full-parameter fine-tuning. For the 8B model, full-parameter tuning significantly harms general knowledge, multilingual, and coding abilities, while Keystone-only keeps these competencies close to Base and occasionally improves them. For the 1B model, both approaches yield gains beyond the Base model, but Keystone-only consistently attains the strongest overall performance. Overall, Keystone-only achieves better math specialization while substantially mitigating degradation on other capability dimensions, an effect that is especially evident in the 8B model.

### 4.3. Safeguarding fine-tuning results

**Keystone-only updates also improve safeguarding performance.** As shown in Table 5, on the WildGuard safety benchmark both fine-tuning strategies improve over the Base models, but Keystone-only consistently achieves the

*Table 5.* WildGuard test accuracy (%) under different fine-tuning settings.

| Model | Setting | WildGuard |
|---|---|---|
| **Llama-3.1-8B-Instruct** | Base | 71.45 |
| | Full FT | 75.65 |
| | Keystone-only | **82.82** |
| **Llama-3.2-1B-Instruct** | Base | 52.03 |
| | Full FT | 69.22 |
| | Keystone-only | **73.40** |

largest gains. For Llama-3.1-8B-Instruct, full-parameter fine-tuning yields a moderate increase in safety accuracy relative to the original checkpoint, while Keystone-only attains a substantially higher score despite updating only a small fraction of parameters. A similar pattern holds for Llama-3.2-1B-Instruct: full-parameter fine-tuning already provides a sizable improvement over Base, yet Keystone-only further raises accuracy and remains the strongest configuration. These results indicate that restricting updates to the keystone-neuron set not only enhances intrinsic model capabilities but also has the potential to improve perfor-

*Table 6.* Mean absolute weight drift of keystone and non-keystone parameters under Full FT and Keystone-only SFT on Math experiment.

| Model | Setting | Keystone | Non-Keystone |
|---|---|---|---|
| **Llama-3.1-8B-Instruct** | Full FT | $2.7 \times 10^{-4}$ | $2.5 \times 10^{-4}$ |
| | Keystone-only | $\mathbf{5.0 \times 10^{-8}}$ | 0 |
| **Llama-3.2-1B-Instruct** | Full FT | $2.3 \times 10^{-4}$ | $2.1 \times 10^{-4}$ |
| | Keystone-only | $\mathbf{3.2 \times 10^{-8}}$ | 0 |

mance on a specialized downstream task, while using far fewer trainable weights than full-parameter fine-tuning.

## 4.4. Discussion

**Weight-drift analysis indicates that Keystone-only performs highly localized adjustments, whereas full fine-tuning induces widespread parameter reconfiguration.** To characterize how each strategy modifies the underlying network, we analyze weight drift, measured as the mean absolute parameter change before versus after fine-tuning, separately for keystone and non-keystone blocks (Table 6). Under Full FT for both Llama-3.1-8B-Instruct and Llama-3.2-1B-Instruct, keystone and non-keystone parameters exhibit comparable drift magnitudes ($2$–$3 \times 10^{-4}$), consistent with broad, model-wide rewriting during math SFT. In contrast, Keystone-only effectively freezes non-keystone parameters (drift numerically indistinguishable from zero) and updates keystone parameters only minimally ($3$–$5 \times 10^{-8}$), approximately four orders of magnitude smaller than Full FT. Since neurons in $\mathcal{S}_{\text{keystone}}$ are persistently highly activated across heterogeneous prompts and their joint ablation collapses performance across benchmarks, they function as a high-throughput backbone rather than narrowly specialized feature units. Restricting updates to this backbone therefore corresponds to making small, structured recalibrations of global information flow—adjusting gains along existing high-traffic channels instead of rewriting task- or language-specific submodules—and acts as an implicit regularizer when fine-tuning on narrow math data. This mechanistic contrast mirrors the evaluation outcomes: unconstrained full-parameter math SFT is more susceptible to representation drift and negative transfer, whereas Keystone-only improves math performance while largely preserving non-math capabilities, and can even amplify gains when Full FT is already mildly beneficial (e.g., on the 1B model).

## 5. Related Work

**Modularity in LLMs.** A growing line of work suggests that LLM computation is organized into structured subcomponents rather than being uniformly distributed. In sparse MoE systems, routing selects a small subset of experts per token; GShard (Lepikhin et al., 2020), Switch Transformers (Fedus et al., 2022), and ST-MoE (Zoph et al., 2022)

report expert- and layer-level specialization patterns that emerge at scale. Modularity also appears in dense Transformers without explicit routing: Qiu et al. recover expert-like functional blocks (Qiu et al., 2024), and Zhang et al. show functionally distinct neuron clusters that behave as implicit experts (Zhang et al., 2023). Related analyses emphasize that capabilities concentrate in particular regions or circuits (Pochinkov, 2023), while sparse/monosemantic decompositions provide an additional route to making such structure more interpretable (Bricken et al., 2023). Beyond post-hoc discovery, modularity can be encouraged or quantified via explicit mechanisms, including domain-tag routing in Demix Layers (Gururangan et al., 2021) and weight-masking based modular analysis (Csordás et al., 2021), echoing sparsely interacting modules studied in recurrent independent mechanisms (Goyal et al., 2020).

**Specialized neurons.** At a finer granularity, many works identify specialized neurons by associating unit activations with an attribute and validating causality through targeted interventions. Tang et al. report language-specific neurons and show that intervening on them selectively perturbs multilingual behavior (Tang et al., 2024); Song et al. identify task-specific neurons with similarly selective effects (Song et al., 2024); and Chen et al. provide evidence for neurons or linear directions encoding abstract cross-lingual factors (Chen et al., 2025). For factual knowledge, Dai et al. propose knowledge neurons tied to specific associations, showing that editing/ablating only these units can flip targeted answers (Dai et al., 2022); related views interpret FFNs as key–value memories in which neuron activations gate fact retrieval (Geva et al., 2021), and ROME enables causal knowledge edits via compact subspace updates (Meng et al., 2023). In parallel, sparsity has been linked to trainability and interpretability through lottery-ticket subnetworks (Frankle and Carbin, 2019; Chen et al., 2020) and monosemantic sparse autoencoder features that map to high-level concepts (Bricken et al., 2023). In contrast to these task-specific units, our work highlights a sparse subset of neurons that remains persistently highly activated across different tasks, suggesting a intrinsic subset spanning diverse capabilities.

## 6. Conclusion

In this study, we find that large language models contain an extremely small but structurally critical set of keystone neurons. Across all evaluated models, we can isolate a subset of neurons that is strongly activated under diverse prompts, occupies only a tiny fraction of all neurons, yet whose joint masking causes the model to effectively lose all of its capabilities. Structurally, these neurons concentrate on a high-activation frontier within attention and feed-forward modules and form a depth-wise backbone that is largely established during pretraining and is mainly thickened by

instruction tuning. Keystone neurons also react much more sharply than random neurons under multiplicative scaling, and restricting supervised fine-tuning to their associated weights already yields substantial and comparatively stable gains with much smaller weight drift than full-parameter updates, which better preserves other intrinsic capabilities.

## Limitations and Future Work

Our work has several limitations and suggests multiple directions for future research. First, due to compute constraints, experiments are limited to open-weight models in the 0.5B–30B range, spanning both dense and MoE architectures; whether similarly sparse and structurally critical keystone backbones persist in much larger dense models, extremely large expert models, or proprietary frontier systems remains open. Second, since both compute and a universally accepted taxonomy of LLM capabilities are lacking, we cannot exhaustively enumerate capability axes or prompts; we therefore follow the capability dimensions evaluated in the Qwen technical report as a practical proxy, which offers broad coverage but does not guarantee convergence to a unique, fully stable keystone neurons subset under arbitrary expansions. Nevertheless, the results in our experiment consistently suggest that keystone neurons represent a broadly observed structural pattern across diverse LLMs.

A natural next step is therefore to extend the analysis to broader and more heterogeneous model families, and to study how keystone neurons interact with model compression and quantization, for example by designing schemes that explicitly preserve or reweight these neurons and examining the resulting robustness and failure modes.

## Impact Statement

This paper presents work whose goal is to advance the field of machine learning by studying a prompt-stable set of keystone neurons in large language models. There are many potential societal consequences of this kind of research, none of which we feel must be specifically highlighted here.

## Acknowledgment

This research is supported by the National Science and Technology Major Project (2023ZD0121102), the National Research Foundation, Singapore, under its National Large Language Models Funding Initiative (AISG Award No: AISG-NMLP-2024-002). Any opinions, findings and conclusions or recommendations expressed in this material are those of the authors and do not reflect the views of National Research Foundation, Singapore.

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

# A. Training Details

All fine-tuning runs share the same optimization setup. We use AdamW with a learning rate of $2 \times 10^{-5}$, weight decay 0.01, global batch size 32, and train for 2 epochs over the 10k math instruction–response pairs. The maximum input sequence length is 2048 tokens. The learning rate schedule uses a warmup ratio of 0.03 followed by linear decay.

For the **Full FT** setting, all model parameters are updated. For the **Keystone-only** setting, we keep the optimizer, learning rate schedule, and batch configuration identical, but enforce sparsity at the gradient level: after back-propagation, the gradients of all non-keystone neurons are set to zero before the optimizer step. Operationally, this masks out all rows or columns that are not in $\mathcal{S}_{\text{keystone}}$, so only keystone-related weights are updated.

## A.1. Math fine-tuning Prompt template and example

**Zero-shot prompt template.**    All math fine-tuning runs use the following zero-shot prompt template:

```
Please reason step by step, and put your final answer within \boxed{}.

This is the problem:

<<Question>>
```

Here, `<<Question>>` is replaced by the raw math question.

**Example prompt with question.**    For example, plugging a quadratic-function problem into the template gives the following full input prompt:

```
Please reason step by step, and put your final answer within \boxed{}.

This is the problem:

Let $g$ be a quadratic function for which $g(3) = g(5)$ and $g(4) = 10$.
What is the value of $g(6) - g(2)$?
```

The corresponding training target is the full chain-of-thought solution ending with a boxed final answer, and during evaluation we always extract the expression inside `\boxed{}` as the final answer for scoring.

### A.2. Safeguarding fine-tuning prompt template

**Zero-shot prompt template.** All safeguarding fine-tuning runs use the following zero-shot prompt template:

```
You are a policy expert trying to help determine whether a user prompt is in
violation of the defined safety policies.
Task:  Check if there is unsafe content in 'User' messages in conversation
according to our safety policy with the below categories.

<BEGIN UNSAFE CONTENT CATEGORIES>
U1 - Sensitive Information (Organization)
U2 - Private Information (Individual)
U3 - Copyright Violations
U4 - False or Misleading Information
U5 - Material Harm by Misinformation
U6 - Social Stereotypes & Discrimination
U7 - Violence and Physical Harm
U8 - Toxic Language / Hate Speech
U9 - Sexual Content
U10 - Cyberattacks
U11 - Fraud & Assisting Illegal Activities
U12 - Encouraging Unethical / Unsafe Actions
U13 - Mental Health & Over-Reliance Crisis
<END UNSAFE CONTENT CATEGORIES>

<BEGIN CONVERSATION>
User:  <<Question>>
<END CONVERSATION>

Your final answer safe or unsafe MUST BE put in \boxed{}.
```

Here, `<<Question>>` is replaced by the raw user prompt to be classified.

The corresponding training target is a single-token classification label (`safe` or `unsafe`) enclosed in `\boxed{}`. During evaluation, we always extract the expression inside `\boxed{}` as the final prediction for scoring.

# B. Prompt groups for keystone detection

For Stage 1 activation analysis, we use small prompt groups that cover four capability dimensions in a balanced way: general knowledge (MMLU), mathematical reasoning (MATH500), code generation (EvalPlus), and multilingual tasks (MGSM). Each group contains one prompt per dimension. Below we list two of the prompt groups used in our experiments.

## B.1. Prompt Group 1

```
[MMLU]
Q: Suppose sharing of tiles in a multilevel directory structure is achieved
with directory entries that are links pointing to a node containing
information about a shared file.  Information in this node includes (1)
the owner of the file, (2) a count of the number of links to the tile, and
(3) the disk block numbers of the file.  What is a primary drawback to this
approach to sharing?
A. If the owner modifies the file, another user who does not share will see
the changes.
B. If the owner renames the file, other users will not be able to access it.
C. If the owner is allowed to delete a file, dangling links may result.
D. If any user who shares the file appends to it, others who share it will
not be able to access the new disk blocks.
Please think briefly if needed, then output strictly in this single line
format:  The final answer is [X].

[MATH500]
Solve the following math problem by reasoning step by step and put your final
answer within \boxed{}.
This is the problem:
What is the distance, in units, between the points (2,−6) and (−4,3) Express
your answer in simplest radical form.

[EvalPlus]
Write a python function to identify non-prime numbers.

[MGSM]
问题：利亚有32 块巧克力，她妹妹有42 块。如果她们吃了35 块，她们一共还剩下多少块？
```

### B.2. Prompt Group 2.

**[MMLU]**
Q: For Socrates, an unexamined life is a tragedy because it results in grievous harm to _____.
A. the state
B. the justice system
C. the body
D. the soul
Please think briefly if needed, then output strictly in this single line format:  The final answer is [X].

**[MATH500]**
Solve the following math problem by reasoning step by step and put your final answer within \boxed{}.
This is the problem:
Let $p(x)$ be a polynomial of degree 5 such that $p(n) = \frac{n}{n^2-1}$ for $n = 2, 3, 4, 5, 6, 7$. Find $p(8)$.

**[EvalPlus]**
Write a function to check if all the elements in a tuple have the same data type or not.

**[MGSM]**
问题：肖恩有五个玩具。圣诞节他从他爸爸妈妈那里各得到了两个玩具。他现在有多少个玩具？

### B.3. Prompt Group 3.

**[MMLU]**
Q: According to Piaget, children are _____.
A. Blank slates
B. Less intelligent than adults
C. Little scientists
D. Shaped by culture
Please think briefly if needed, then output strictly in this single line format:  The final answer is [X].

**[MATH500]**
Solve the following math problem by reasoning step by step and put your final answer within \boxed{}.
This is the problem:
Find the greatest integer less than $(\sqrt{7} + \sqrt{5})^6$.

**[EvalPlus]**
Write a function to remove characters from the first string which are present in the second string.

**[MGSM]**
问题：杰森有20 根棒棒糖。他给了丹尼一些棒棒糖。现在杰森有12 根棒棒糖。杰森给了丹尼多少根棒棒糖？

## B.4. Prompt Group 4.

**[MMLU]**
Q: Identify the antecedent of the following conditional proposition: If the university does not increase financial aid, either the president fails to approve it or the board of trustees prevents it.
A. The university increases financial aid.
B. The university does not increase financial aid.
C. The board of trustees prevents it.
D. The president fails to approve it.
Please think briefly if needed, then output strictly in this single line format: The final answer is [X].

**[MATH500]**
Solve the following math problem by reasoning step by step and put your final answer within \boxed{}.
This is the problem:
Find the largest value of $x$ that satisfies the equation $|5x-1| = x+3$.

**[EvalPlus]**
Write a python function to check whether the given number can be represented as the difference of two squares or not.

**[MGSM]**
问题：迈克尔有58 个高尔夫球。周二，他丢失了23 个高尔夫球。周三，他又丢失了2 个。周三结束时他有多少个高尔夫球？

## B.5. Prompt Group 5.

**[MMLU]**
Q: Aesthetics deals with objects that are ____.
A. essential to our existence
B. unimportant to most people
C. not essential to our existence
D. rarely viewed
Please think briefly if needed, then output strictly in this single line format: The final answer is [X].

**[MATH500]**
Solve the following math problem by reasoning step by step and put your final answer within \boxed{}.
This is the problem:
Convert the point (0, 3) in rectangular coordinates to polar coordinates.
Enter your answer in the form (r, theta), where r > 0 and 0 <= theta < 2 pi.

**[EvalPlus]**
Write a function to find the sum of numbers in a list within a range specified by two indices.

**[MGSM]**
问题：服务器机房里有九台电脑。从周一到周四，每天又安装了五台电脑。服务器机房里现在有多少台电脑？

# C. Additional Experimental Results

## C.1. Full keystone-ablation table

Table 7 and Table 8 report the complete ablation results for all models under the **Base**, **Keystone-off**, and **Random-off** settings (accuracy higher is better, perplexity lower is better).

*Table 7.* Full neuron-masking results (Qwen3-dense, Gemma3, and Llama3 series). The first four columns are accuracies on evaluation benchmarks; the last two columns are perplexities on C4 and WikiText-2.

| Model / setting | MMLU | Math500 | MGSM | EvalPlus | C4 | W2 |
|---|---|---|---|---|---|---|
| **Qwen3-dense series** | | | | | | |
| Qwen3-0.6B | 0.53 | 0.72 | 0.684 | 0.423 | 29.97 | 33.73 |
| Qwen3-0.6B (Keystone-off) | 0 | 0 | 0 | 0 | 340 | 566 |
| Qwen3-0.6B (Random-off) | 0.518 | 0.54 | 0.616 | 0.431 | 32.28 | 39.37 |
| Qwen3-8B | 0.821 | 0.942 | 0.842 | 0.669 | 15.74 | 15.69 |
| Qwen3-8B (Keystone-off) | 0 | 0 | 0 | 0 | 19.53 | 23.74 |
| Qwen3-8B (Random-off) | 0.824 | 0.638 | 0.856 | 0.651 | 17.15 | 17.43 |
| **Gemma3 series** | | | | | | |
| Gemma-3-1B-IT | 0.387 | 0.438 | 0.468 | 0.386 | 297 | 299 |
| Gemma-3-1B-IT (Keystone-off) | 0 | 0 | 0 | 0 | 50938 | 130328 |
| Gemma-3-1B-IT (Random-off) | 0.373 | 0.440 | 0.462 | 0.386 | 255.31 | 114.76 |
| Gemma-3-1B-PT | 0.316 | 0.400 | 0.410 | 0.362 | 313 | 347 |
| Gemma-3-1B-PT (Keystone-off) | 0 | 0 | 0 | 0 | 52927 | 113547 |
| Gemma-3-1B-PT (Random-off) | 0.302 | 0.386 | 0.400 | 0.358 | 3069.37 | 123.17 |
| **Llama3 series** | | | | | | |
| Llama-3.2-1B-Instruct | 0.410 | 0.236 | 0.300 | 0.246 | 11.83 | 17.77 |
| Llama-3.2-1B-Instruct (Keystone-off) | 0 | 0 | 0 | 0 | 779 | 807 |
| Llama-3.2-1B-Instruct (Random-off) | 0.401 | 0.204 | 0.262 | 0.235 | 21.01 | 24.52 |
| Llama-3.2-1B | 0.328 | 0.205 | 0.240 | 0.224 | 13.32 | 19.06 |
| Llama-3.2-1B (Keystone-off) | 0 | 0 | 0 | 0 | 1770 | 4087 |
| Llama-3.2-1B (Random-off) | 0.319 | 0.194 | 0.232 | 0.186 | 13.84 | 16.94 |
| Llama-3.1-8B-Instruct | 0.713 | 0.496 | 0.792 | 0.532 | 11.78 | 13.61 |
| Llama-3.1-8B-Instruct (Keystone-off) | 0 | 0 | 0 | 0 | 522 | 879 |
| Llama-3.1-8B-Instruct (Random-off) | 0.706 | 0.486 | 0.770 | 0.545 | 11.83 | 17.76 |
| Llama-3.1-8B | 0.627 | 0.450 | 0.696 | 0.482 | 9.70 | 9.59 |
| Llama-3.1-8B (Keystone-off) | 0 | 0 | 0 | 0 | 1939 | 2011 |
| Llama-3.1-8B (Random-off) | 0.604 | 0.446 | 0.690 | 0.478 | 13.31 | 11.89 |

*Table 8.* Full ablation results (Qwen2.5 series, Reasoning-distilled models, and MoE models). The first four columns are accuracies on evaluation benchmarks; the last two columns are perplexities on C4 and WikiText-2.

| Model / setting | MMLU | Math500 | MGSM | EvalPlus | C4 | W2 |
|---|---|---|---|---|---|---|
| **Qwen2.5 series** | | | | | | |
| Qwen2.5-0.5B-Instruct | 0.355 | 0.316 | 0.410 | 0.388 | 21.22 | 23.15 |
| Qwen2.5-0.5B-Instruct (Keystone-off) | 0 | 0 | 0 | 0 | 52.52 | 65.72 |
| Qwen2.5-0.5B-Instruct (Random-off) | 0.347 | 0.282 | 0.405 | 0.391 | 22.52 | 25.64 |
| Qwen2.5-0.5B | 0.320 | 0.276 | 0.392 | 0.368 | 31.94 | 19.79 |
| Qwen2.5-0.5B (Keystone-off) | 0 | 0 | 0 | 0 | 36.95 | 48.55 |
| Qwen2.5-0.5B (Random-off) | 0.312 | 0.258 | 0.375 | 0.353 | 21.23 | 23.82 |
| Qwen2.5-7B-Instruct | 0.757 | 0.748 | 0.870 | 0.632 | 12.42 | 12.56 |
| Qwen2.5-7B-Instruct (Keystone-off) | 0 | 0 | 0 | 0 | 18824 | 561 |
| Qwen2.5-7B-Instruct (Random-off) | 0.745 | 0.688 | 0.840 | 0.627 | 13.83 | 14.52 |
| Qwen2.5-7B | 0.702 | 0.700 | 0.812 | 0.605 | 11.71 | 12.01 |
| Qwen2.5-7B (Keystone-off) | 0 | 0 | 0 | 0 | 115 | 79.8 |
| Qwen2.5-7B (Random-off) | 0.674 | 0.692 | 0.805 | 0.598 | 12.78 | 13.12 |
| **Reasoning-distilled models** | | | | | | |
| DeepSeek-R1-Distill-Llama-8B | 0.711 | 0.848 | 0.762 | 0.537 | 33.31 | 27.18 |
| DeepSeek-R1-Distill-Llama-8B (Keystone-off) | 0 | 0 | 0 | 0 | 20981 | 31526 |
| DeepSeek-R1-Distill-Llama-8B (Random-off) | 0.690 | 0.680 | 0.756 | 0.513 | 189 | 63.84 |
| DeepSeek-R1-Distill-Qwen-1.5B | 0.477 | 0.816 | 0.694 | 0.410 | 69.17 | 72.84 |
| DeepSeek-R1-Distill-Qwen-1.5B (Keystone-off) | 0 | 0 | 0 | 0 | 521 | 391 |
| DeepSeek-R1-Distill-Qwen-1.5B (Random-off) | 0.458 | 0.586 | 0.608 | 0.410 | 500.86 | 219.44 |
| DeepSeek-R1-Distill-Qwen-7B | 0.654 | 0.892 | 0.848 | 0.563 | 27.26 | 20.10 |
| DeepSeek-R1-Distill-Qwen-7B (Keystone-off) | 0 | 0 | 0 | 0 | 7068 | 7781 |
| DeepSeek-R1-Distill-Qwen-7B (Random-off) | 0.645 | 0.752 | 0.798 | 0.547 | 417.21 | 141.05 |
| **MoE models** | | | | | | |
| Qwen3-30B-A3B-Instruct | 0.865 | 0.972 | 0.930 | 0.698 | 14.77 | 12.10 |
| Qwen3-30B-A3B-Instruct (Keystone-off) | 0 | 0 | 0 | 0 | 206.91 | 152.10 |
| Qwen3-30B-A3B-Instruct (Random-off) | 0.842 | 0.826 | 0.854 | 0.669 | 14.46 | 12.84 |
| Qwen3-30B-A3B | 0.857 | 0.946 | 0.918 | 0.648 | 16.35 | 15.35 |
| Qwen3-30B-A3B (Keystone-off) | 0 | 0 | 0 | 0 | 282 | 129 |
| Qwen3-30B-A3B (Random-off) | 0.841 | 0.580 | 0.880 | 0.632 | 16.22 | 16.50 |
| Mixtral-8x7B-v0.1-Instruct | 0.645 | 0.284 | 0.528 | 0.476 | 32.12 | 11.14 |
| Mixtral-8x7B-v0.1-Instruct (Keystone-off) | 0 | 0 | 0 | 0 | 580 | 250 |
| Mixtral-8x7B-v0.1-Instruct (Random-off) | 0.609 | 0.248 | 0.490 | 0.402 | 47.35 | 12.90 |
| Mixtral-8x7B-v0.1 | 0.538 | 0.268 | 0.517 | 0.460 | 40.32 | 12.25 |
| Mixtral-8x7B-v0.1 (Keystone-off) | 0 | 0 | 0 | 0 | 9445 | 6807 |
| Mixtral-8x7B-v0.1 (Random-off) | 0.526 | 0.246 | 0.472 | 0.400 | 25.35 | 9.81 |

## C.2. Activation patching

Table 9 reports the full activation-patching results for Llama-3.1-8B-Instruct. For each source capability, keystone-neuron activations are replaced by either the response-level mean activation or the activation at the first generated token computed from that source prompt.

*Table 9.* Full activation-patching results for keystone neurons.

| Patch statistic | Source capability | MMLU | Math500 | MGSM | EvalPlus |
|---|---|---|---|---|---|
| Native activations | – | 0.713 | 0.496 | 0.792 | 0.532 |
| Response mean | General tasks | 0 | 0 | 0 | 0 |
| | Mathematics | 0 | 0 | 0 | 0 |
| | Code generation | 0 | 0 | 0 | 0 |
| | Multilingual tasks | 0 | 0 | 0 | 0 |
| First token | General tasks | 0 | 0 | 0 | 0 |
| | Mathematics | 0 | 0 | 0 | 0 |
| | Code generation | 0 | 0 | 0 | 0 |
| | Multilingual tasks | 0 | 0 | 0 | 0 |

## C.3. Additional controls

We provide stronger controls for the ablation results to test whether the observed collapse can be explained by weaker alternatives such as module location, parameter magnitude, random ablation budget, or the use of zero masking itself.

**Module-aware random ablation.** To test whether the effect simply arises from ablating neurons in especially important modules, we randomly deactivate the same number of neurons within each individual projection type, including FFN up/down projections and attention $Q/K/V/O$ projections. Table 10 shows that these module-aware random interventions cause only limited degradation and remain far weaker than keystone-neuron ablation.

*Table 10.* Module-aware random ablation results.

| Model / module | MMLU | Math500 | MGSM | EvalPlus |
|---|---|---|---|---|
| Llama-3.1-8B-Instruct | 0.713 | 0.496 | 0.792 | 0.532 |
| FFN up | 0.716 | 0.450 | 0.796 | 0.545 |
| FFN down | 0.706 | 0.484 | 0.784 | 0.545 |
| Attention $Q$ | 0.713 | 0.486 | 0.788 | 0.548 |
| Attention $K$ | 0.702 | 0.482 | 0.764 | 0.526 |
| Attention $V$ | 0.715 | 0.498 | 0.790 | 0.542 |
| Attention $O$ | 0.718 | 0.482 | 0.796 | 0.548 |
| Qwen2.5-7B-Instruct | 0.757 | 0.748 | 0.870 | 0.632 |
| FFN up | 0.795 | 0.764 | 0.872 | 0.619 |
| FFN down | 0.790 | 0.740 | 0.862 | 0.619 |
| Attention $Q$ | 0.774 | 0.760 | 0.874 | 0.619 |
| Attention $K$ | 0.758 | 0.628 | 0.802 | 0.556 |
| Attention $V$ | 0.740 | 0.692 | 0.840 | 0.571 |
| Attention $O$ | 0.788 | 0.750 | 0.874 | 0.609 |

**High-norm neuron ablation.** To test whether keystone neurons are important merely because they have large parameter magnitudes, we ablate neurons selected by parameter norm under the same neuron budget. As shown in Table 11, high-norm ablation degrades performance but does not reproduce the complete collapse caused by keystone-neuron ablation.

*Table 11.* High-norm neuron ablation results.

| Model / setting | MMLU | Math500 | MGSM | EvalPlus | C4 | W2 |
|---|---|---|---|---|---|---|
| Llama-3.1-8B-Instruct | 0.713 | 0.496 | 0.792 | 0.532 | 11.78 | 13.61 |
| High-norm neurons | $0.676 \pm 0.041$ | $0.295 \pm 0.105$ | $0.645 \pm 0.064$ | $0.443 \pm 0.043$ | $13.60 \pm 0.62$ | $16.07 \pm 1.73$ |
| Qwen2.5-7B-Instruct | 0.757 | 0.748 | 0.870 | 0.632 | 12.42 | 12.56 |
| High-norm neurons | $0.452 \pm 0.030$ | $0.108 \pm 0.007$ | $0.324 \pm 0.043$ | $0.200 \pm 0.026$ | $14.55 \pm 0.01$ | $16.52 \pm 0.09$ |

**Random-neuron ablation sweep.** We further ask how many randomly selected neurons must be removed before random ablation reaches a collapse comparable to keystone-neuron masking. Table 12 shows that Llama-3.1-8B-Instruct requires ablating a much larger random subset, approaching $10\%$ of all neurons, before all four benchmark accuracies drop to zero.

*Table 12.* Random-neuron ablation sweep on Llama-3.1-8B-Instruct.

| Random ratio (count) | MMLU | Math500 | MGSM | EvalPlus | C4 | W2 |
|---|---|---|---|---|---|---|
| Base | 0.713 | 0.496 | 0.792 | 0.532 | 11.78 | 13.61 |
| 0.0001 (64) | 0.739 | 0.510 | 0.784 | 0.542 | 11.66 | 13.46 |
| 0.001 (1,408) | 0.705 | 0.478 | 0.802 | 0.537 | 11.98 | 13.89 |
| 0.003 (4,288) | 0.698 | 0.404 | 0.666 | 0.497 | 12.32 | 14.28 |
| 0.006 (8,576) | 0.579 | 0.220 | 0.480 | 0.320 | 14.86 | 18.25 |
| 0.008 (11,392) | 0.518 | 0.106 | 0.340 | 0.349 | 15.55 | 18.77 |
| 0.01 (14,272) | 0.497 | 0.130 | 0.398 | 0.299 | 19.02 | 24.08 |
| 0.03 (43,136) | 0.214 | 0 | 0.012 | 0 | 142.29 | 249.76 |
| 0.05 (71,936) | 0.186 | 0 | 0.020 | 0 | 447.36 | 658.63 |
| 0.07 (100,800) | 0.044 | 0 | 0 | 0 | 1647.06 | 3256.93 |
| 0.10 (144,064) | 0 | 0 | 0 | 0 | 7910.34 | 15948.83 |

**Same-layer same-module random replacement.** Finally, to test whether the main result is specific to zero masking, we replace each keystone neuron with a randomly selected neuron from the same layer and projection module. Table 13 shows that this nonzero replacement-based intervention also causes complete collapse, indicating that the specific keystone-neuron parameter configuration is not interchangeable with ordinary neurons from the same local context.

*Table 13.* Same-layer same-module random replacement results.

| Model / setting | MMLU | Math500 | MGSM | EvalPlus | C4 | W2 |
|---|---|---|---|---|---|---|
| Llama-3.1-8B-Instruct | 0.713 | 0.496 | 0.792 | 0.532 | 11.78 | 13.61 |
| Random replacement | $0 \pm 0$ | $0 \pm 0$ | $0 \pm 0$ | $0 \pm 0$ | $3647.47 \pm 1452.48$ | $4333.61 \pm 952.72$ |
| Qwen2.5-7B-Instruct | 0.757 | 0.748 | 0.870 | 0.632 | 12.42 | 12.56 |
| Random replacement | $0 \pm 0$ | $0 \pm 0$ | $0 \pm 0$ | $0 \pm 0$ | $12755.51 \pm 8526.60$ | $6596.78 \pm 5899.67$ |

