# OpenReview forum: "Tiny Brains, Giant Impact: Uncovering the Keystone Neurons of LLM with Just a Few Prompts"
_ICML.cc/2026/Conference — ICML 2026 regular_

### Official Review · Reviewer_VRVt · 2026-03-05

**Soundness:** 3
**Presentation:** 4
**Significance:** 2
**Originality:** 2
**Overall Recommendation:** 4
**Confidence:** 5

**Summary:**

This paper proposes a method for identifying keystone neurones in LLMs, an extremely sparse set of neurones that, when ablated, consistently lead to a dramatic drop of performance in many tasks. The proposed method is data efficient, as it merely requires four prompts. Building on this finding, the paper propose a parameter-efficient fine-tuning method consisting in only updating the keystone neurones.
Although the paper is well written and the narrative is appealing, it suffers from severe limitations. Adding a more rigorous set of control experiments, as well as revising the narrative of the paper should be a pre-requisite for being published at the conference.

**Compliance With Llm Reviewing Policy:**

Affirmed.

**Final Justification:**

In summary, I am grateful to the authors for writing this rebuttal. They have provided additional explanations and experiments that address my main concerns. I am therefore raising my rating to "acceptable" (with reservations). In my opinion, the remaining limitations are: (1) the poor quality of the full model baseline fine-tuning weakens the results of the keystone fine-tuning; (2) the impact of the study remains limited because it only discovers a set of indispensable neurons, without any evidence of the role they play during processing (for example, these neurons are far from sufficient to process the input).

**Key Questions For Authors:**

1. Are you including the FFN gates when looking for keystone neurons? And embedding and de-embedding matrices? I understand that no, but the authors could clarify it. Although I understand why authors did not consider the embedding and de-embedding; I’m surprised that they ignored the gate projection.

2. “keystone neurons are largely established during pre-training” (Section 3.5): it could be interesting to know at which stage of the training they emerge, have you looked at it? (I’m not saying this should be done, I’m just curious)

3. Where are located the keystone neurones more precisely? Could provide the number of keystone neurones per-layer and per modules please?

4. Could you remind how the scores in table 1 are computed please? The perfect 0 obtained with all keystone-off ablation is surprising.

5. Clarification: in table 3, do you confirm that the models are fine-tuned on the math data, and then evaluated on the non-math benchmark? I.e. they are never trained on the non-math data, right?

6. Fig 2: could you add uncertainty estimations, such as the std for each bar plot and accuracy line plot, please?

**Limitations:**

For the technical limitations, refer to my evaluation of the soundess.

**Strengths And Weaknesses:**

I detail hereby my evaluation of the soundness, significance, presentation and originality of the paper. I note with “+” the strengths and with “-” the weakness.

**Soundness (2/4):**

** *[edit] -> Increased to 3/4, cf. rebuttal and comments.* **

\+ the method is efficient, allowing to identify keystone neurones with only four prompts.

\+ Fig 3,4,5 provide interesting insight on the nature of this critical set of neurons, related to their stability and their distribution across layers and across models.

\- This paper makes strong claims about the sparsity (“Tiny Brains, Giant Impact”) without making the proper control experiments that support these claims. This is particularly important as LLMs are highly distributed system *by design*, thus claiming an extreme sparsity is very interesting but should be rigorously demonstrated. In the “significance” section I’ll discuss why I think the main claim about sparsity is incorrect and misleading. Here, I’ll only focus on the technical aspect, and describe the lack of proper baseline/ablation/control experiments:

*A) The random baseline:* Authors compare the ablation of keystone neurons with the ablation of a same amount of random neurons, showing that the latter does not reproduce the drop in accuracy obtained with the former. This kind of ablation is essential to understand the importance of the keystone neurons. However, I believe that this random baseline is too weak.

  A-1) It would be informative to try prior-informed random baseline. For instance, if the keystone neurones are disproportionately distributed on the MLPs of the last layers, it would makes sense to also try a stronger random baseline that randomly ablate neurones from these modules only. Alternatively, if the keystones are more present in the attention (which seems to be the case, based on Fig. 4), it could be informative to also try removing random neurones from the attention only. In addition, other baselines can be considered, for instance ablating the neurones which have the higher norm in the parameters, regardless of the activation.

A-2) The current random baseline comparison could also be more informative if it showed how many random neurons one must ablate to observe the same degradation as for keystone neurones. Indeed, authors showed a sharp drop with keystones neurones, maybe the drop is also sharp for random ones, and ablating only a few more random neurons could lead to a drop in performance.

  *B) The fine-tuning:* Authors showed that fine-tuning only the keystone neurons leads to better performance than fine-tuning the full model.

B-1) More precisely, Table 3 shows that in almost all the tasks considered, *the full-fine tuning degrades the task performance* compared to the non-fine-tuned baseline, while the keystone fine-tune slightly improve the performance. In my opinion, the fact that full finetuning degrades the performance suggests that the authors are not doing it properly. Did the authors conduct separate grid searches to find the best hyper-parameters for fine-tuning? The full finetuning score should at least be equal to the baseline one, e.g. if using a very small learning rate. I might be wrong, but the authors should at least provide the full description of the fine-tuning procedure, including the hyper-parameter search, otherwise we cannot asses the validity of this experiment. Furthermore, it does not makes sense to use the same set of hyper parameters between the Full FT and the keystone-only settings, as the two setup are very different (e.g. updating ~100 keystone neurones vs. the 1,00,000 neurones that compose the full model).

B-2) In addition, a fair control baseline would consist in fine-tuning a random set of neurones which size is equal to the keystone one. And as previously suggested, measuring the number of random neurons one has to fine-tune to reach the keystone fine-tuning performance.

*C) The neuron ablation:*

  C-1) To measure the impact of removing the keystone neurons on the LLM performance, the authors set the corresponding activation to zero. The literature seems to suggest that zero-ing activation is misleading as it can severely distort the in-distribution of the hidden representation [c], leading to an over-estimation of the pruning effect. I think the authors should at least discuss discussing it in the limitation, and eventually try other alternatives although it might not be as trivial as it seems. Maybe they could replace the key-stone neurone by random neurones in the same block? (I’m not convinced my-self).

[c] Li, M. and Janson, L. Optimal ablation for interpretability. NeurIPS, 2024

C-2) Furthermore, to assess how critical are the keystone neurones for the model, it would be interesting to continue few steps of retraining while masking them. If the model can ”repair” the loss of the keystone, we can conclude that, after all, there were not that important. On the contrary, if the loss cannot be compensated, then these neurones are indeed critical. Without such experiment, it is difficult to truly understand how critical they are.

**Significance (2/4):**

\+ the code is available, enabling reproduction.

\- This papers evidenced the presence of keystone neurones, which are necessary for the good working of the LLMs. This finding, while not novel (cf. the related work section of the paper), is efficiently and systematically observed in various LLMs family, demonstrating its robustness. However, I wonder how significant is this finding.

To illustrate my doubts, I’ll start with an (exaggerated) example. Following the reasoning of this paper, one could ablate the unembedding matrix of the LLM and, observing a dramatic drop in performance, could claim that the LLM is actually super sparse and that the most critical operation are performed in the unembedding operation. I think we can reasonably say that this reasoning is false: although this operation is *necessary*, it is far from being *sufficient*. Saying that the unembedding matrix is critical is obvious, but it does not improve our understanding of the LLM.

The experiments conducted in this paper are more subtle, but the reasoning is the same. Authors showed that an extremely small set of neurones is critical for the good performance of the LLMs across many tasks, and that ablating them strongly degrade the performance. However, we do not have any evidence showing that these neurones are sufficient for solving the tasks, and we can reasonably think that they are not. As a consequence, I wonder how useful this finding is?

The authors conducted targeted fine-tuning on keystone neurons, which goal might be to demonstrate that the keystone neurones are essential for the core capabilities of the LLM. However, as previously commented in the soundness section, I think this experiment has strong limitations.

In my opinion, the authors jumped very fast to the conclusion when they link the existence of a sparse set of necessary neurones (the keystone ones) to the existence of a very sparse processing in the LLM. At least, this is what I understand when the paper and the title suggest that this small set of neurone acts as a “tiny brain”. First, I don’t think that the comparison with the brain is relevant here. Second, being a “tiny brain” would requires some degree of sufficiency, which is absolutely not discussed in the paper. Therefore, the main message of the paper is not only false, but also misleading: it gives the false impression that the LLMs can be reduced to an extremely sparse set of neurones. The existing literature did find sparsity in LLMs, but at a much lesser degree: the portion of the model that is *sufficient* to avoid severe degradation of the performance lies around 20% and 50% of the full model size depending on the technique (e.g. involving re-training or not) and the task (more difficult tasks requiring a larger portion of the model), e.g. [a, b].

[a] Liu et al. Deja vu: Contextual sparsity for efficient llms at inference time. In ICML 2023.
[b] Lee et al. Cats: Contextually-aware thresholding for sparsity in large language models. arXiv preprint 2024.

**Presentation (4/4):**

\+ the paper is very clear and well written

\- (Minor) All figures are blurry, making them barely readable, but this could be easily fixed.

**Originality (2/4):**

\- As shown in the related work section, identifying sparse subset of neurones that are critical for the LLM processing is not new. The novelty of this paper might be the method, that allows to find this set more efficiently, with only a small set of prompts.

**Overall Recommendation:**
Because of the strong limitations of this paper, mostly detailed in the soundness and significance sections, I recommend a major revision (2/6). I’ll engage in the discussion with the authors and the other reviewers, and revise my review if needed.

---

> ### Author Rebuttal · Authors · 2026-03-31
>
> W1. The random baseline seems too weak. Can stronger baselines be considered?
> Thank you for these helpful suggestions. We added three stronger controls. Table A1 shows that module-aware random ablation within FFN up/down and attention Q/K/V/O causes only limited degradation and remains far weaker than keystone ablation. Table A2 shows that a high-norm baseline is stronger than plain random ablation but still much weaker than keystone ablation; averaged over 5 seeds, Llama-3.1-8B-Instruct retains 67.58/29.52/44.29/64.48 and Qwen2.5-7B-Instruct retains 45.19/10.80/20.00/32.40 on MMLU/Math500/EvalPlus/MGSM. Table A3 shows that comparable collapse requires a much larger random ablation ratio; for Llama-3.1-8B-Instruct, near-complete collapse appears only around 10%, orders of magnitude larger than the keystone set.
>
> W2. Why does full fine-tuning degrade performance?
> Thank you for raising this important possibility. We tuned learning rates and reported the best-performing configuration for each method; More importantly, this phenomenon is not unique to our setup: recent work reports that fine-tuning instruction-tuned models can degrade reasoning or cross-domain performance[a,b], and attributes part of this effect to distribution shift and changes in internal reasoning behavior rather than simply failed optimization[a,c]. We used the same recipe family as prior OpenMathInstruct2-style settings [d], and also observe normal gains for Llama-3.2-1B-Instruct on math and for both models on the other downstream task. Thus hyperparameter sensitivity may contribute, but is unlikely to fully explain the pattern.
> [a] Clear Minds Think Alike: What Makes LLM Fine-tuning Robust? A Study of Token Perplexity
> [b] On the Generalization of SFT: A Reinforcement Learning Perspective with Reward Rectification
> [c] On the Impact of Fine-tuning on Chain-of-Thought Reasoning
> [d] OpenMathInstruct-2: Accelerating AI for Math with Massive Open-Source Instruction Data
>
> W3. A fair control would be to fine-tune a random neuron set of the same size as the keystone set.
> Thank you for this valuable suggestion. We added exactly this control; see Table B1. On Llama-3.1-8B-Instruct, keystone-only tuning improves Math500/GSM8K/AIME-24 from 49.6/83.40/3.33 to 51.20/84.31/6.67, whereas random-only tuning gives 46.2/81.8/0. This suggests that the gain depends on which neurons are updated, not merely on updating few parameters.
>
> W4. Zeroing activations may distort the hidden-state distribution?
> Thank you for this important point. We agree that zero ablation should be discussed as a limitation. We therefore added a replacement-based intervention in which each keystone neuron is replaced by a random neuron from the same layer and module. As summarized in Table C1, this still causes complete collapse: all benchmark accuracies drop to 0 for both Llama-3.1-8B-Instruct and Qwen2.5-7B-Instruct, while perplexity rises sharply. So the result is not solely an artifact of zeroing.
>
> W5. Can the model repair the loss of keystone neurons under further training?
> Thank you for this helpful suggestion. We performed the requested experiment by continuing training with keystone neurons masked; see Table C2. The model shows partial recovery, but remains clearly worse than keystone-only fine-tuning. On Llama-3.1-8B-Instruct, mask-keystone reaches 24.8/67.4/6.52/0.0 on Math500/GSM8K/Olympiad/AIME-24, versus 51.20/84.31/16.74/6.67 for keystone-only. We therefore interpret keystone neurons as critical to the original model state, while also acknowledging some plasticity.
>
> Q1. Are FFN gate projections included? Why is gate projection omitted?
> Embeddings and de-embeddings are not included. We also do not track the gate projection separately, because the propagated FFN signal is SiLU(gate(x))×up(x) before the down projection; our fwd_down activation already captures this joint effect.
>
> Q2. Where exactly are the keystone neurons located?
> Yes. We will include the full per-layer and per-module listing in the revision/supplementary. Representative examples from Llama-3.1-8B-Instruct include fwd_up(16,13860) and fwd_down(9,4255).
>
> Q3. How are the scores in Table 1 computed?
> Table 1 is computed by evaluating the ablated model on the full test sets of MMLU,Math500,MGSM,and EvalPlus using the same pipeline as the unablated model. The all-zero results arise because keystone-off generations typically degenerate into repetitive loops or garbled outputs. We repeated the analysis with multiple prompt groups (Table 2) and observed the same collapse.
>
> Q4. Are the models trained only on math data and then evaluated on non-math benchmarks?
> Yes.
>
> Q5. Can Figure 2 include uncertainty estimates?
> Yes. We also repeated the analysis using five prompt groups and observed highly consistent trends. We will add variability information in the revision.
>
> Full tables for Table A1/A2/A3/B1/C1/C2 are provided in the linked appendix: https://anonymous.4open.science/r/keystone-48CE/table.md

---

> > ### Author Rebuttal · Reviewer_VRVt · 2026-04-02
> >
> > Thank you for the large rebuttal that addresses many of my concerns regarding the technical soundness: stronger control; alternative to zero ablation; and the plasticity experiment. I think these new experiments can improve the paper and should be included in the new version.
> >
> > However, I’m still worried by the methodology employed in the fine-tuning experiment.
> >
> > 1. If an hyper parameter search has been conducted, I think that the fine-tuning should at least lead to the same performance as the non-finetuning version: for instance by choosing a very low learning rate, or a very small number of optimisation steps. In the current version of the paper, the full fine-tuning degrades the performance. In my opinion this indicates a problem in the hyper parameter search. Could you share the hyper-parameter search with us please?
> >
> > 2. I saw in the other comment that the authors selected the best-performing configuration, has it been done on a separate validation set? Here again, sharing the hyper parameter search could be useful for the reader, and clarify how significant is the finetuning experiment.
> >
> > 3. I understand that the literature already noticed how inefficient is vanilla fine-tuning for LLMs. Therefore, if LLM fine-tuning is so bad, maybe it would be more interesting to opt for a different post-training method that works with the full LLM and then show that keystone finetuning is more powerful.
> >
> > In addition, the rebuttal does not address my main criticism: this study demonstrates the existence of a very small set of neurones that are critical for the LM to generate consistent output, but it does not address how necessary these neurones are. This is particularly problematic because the narrative of the paper tells us that these keystone neurones are like a "tiny brain" (cf. the title) of the LLM; implicilty suggesting that they are responsible for important processing in the model. I do not think that the experiments conducted in the current paper are showing this. In my opinion, this negatively impact both the originality and significance of the current work.
> >
> > As a consequence, I'll increase my recommandation to "weak reject" because the paper is more technically sound that what I initially wrote in my review. However, I still ask for rejection because of my concerns regarding the fine-tuning experiment and the unverified claims that keystone neurones are like a "tiny brain", with all that it implies.

---

> > > ### Author Response · Authors · 2026-04-04
> > >
> > > ## For fine-tuning methodology
> > >
> > > Thank you for this helpful comment. We agree that the fine-tuning setup and model-selection protocol should be stated more clearly.
> > >
> > > We used the same optimization setup for both **Full FT** and **Keystone-only FT**, varying only the learning rate over **2e-5, 1e-5, and 1e-6**. The remaining hyper-parameters were fixed to **batch size = 32**, **weight decay = 1e-2**, and **epochs = 2**. This setup follows the training recipe used in the original paper we build on. We note, however, that their positive result was reported on **Llama-3.1-8B Base**, while our experiments are conducted on **Llama-3.1-8B Instruct**, so the same setup does not guarantee monotonic gains for full-parameter tuning.
> > > For each setting, we held out **1,000 in-distribution examples** as a validation set and selected the **best checkpoint on this validation set** for final evaluation.
> > >
> > > We agree that a smaller learning rate is intuitively helpful for reducing forgetting. However, when full-model updates are repeated over data whose distribution differs from the model’s existing instruction-tuning distribution, smaller per-step updates do not necessarily prevent cumulative representation drift. Consistent with this, our current results do **not** show a clear monotonic relationship between smaller learning rate and less forgetting.
> > >
> > > Because of compute constraints, our current search is limited to these three learning rates rather than being exhaustive.
> > >
> > > ### Learning-rate comparison on Llama-3.1-8B-Instruct
> > >
> > > | Model | LR | Setting | Math500 | GSM8K | Olympiad | AIME-24 | MMLU | EvalPlus | MGSM |
> > > |:-----:|:--:|:-------:|:-------:|:-----:|:--------:|:-------:|:----:|:--------:|:----:|
> > > | Llama-3.1-8B-Instruct | – | **Base** | 49.60 | 83.40 | 16.59 | 3.33 | **71.3** | 53.2 | **77.2** |
> > > | -| 2e-5 | Full FT | 31.40 | 70.05 | 8.15 | 0.00 | 62.5 | 32.2 | 60.8 |
> > > | - | 2e-5 | **Keystone-only** | **51.20** | **84.31** | **16.74** | **6.67** | 69.8 | 55.0 | 76.6 |
> > > | - | 1e-5 | Full FT | 30.80 | 69.20 | 7.80 | 0.00 | 61.8 | 34.8 | 60.4 |
> > > | - | 1e-5 | **Keystone-only** | 50.55 | 83.60 | 16.20 | **6.67** | 70.1 | 56.6 | 76.0 |
> > > | - | 1e-6 | Full FT | 29.60 | 72.80 | 9.38 | 0.00 | 62.0 | 32.8 | 62.4 |
> > > |-| 1e-6 | **Keystone-only** | 48.00 | 84.20 | **16.74** | 3.33 | 70.92 | **55.8** | 76.8 |
> > >
> > > ## For whether keystone neurons necessary
> > > Thank you for raising this concern. Our use of the metaphor “tiny brain” is intended to emphasize the **indispensability** of keystone neurons, in the sense that successful generation appears to require their participation across diverse tasks. In this sense, the analogy is similar to how many human activities require the brain’s involvement, even though the brain alone does not perform every part of the activity in isolation. Likewise, our claim is not that keystone neurons by themselves carry out all reasoning, but rather that they form a small set whose participation is necessary for the model to perform coherent inference. In particular, the new experiment shows that the model depends on their **precise activation values**, rather than merely on their presence or on a large constant contribution.
> > >
> > > To test this directly, we performed an additional **activation patching** experiment on the keystone neurons of **Llama-3.1-8B-Instruct**. For each capability group (**general-task**, **math**, **code**, **multilingual**), we replaced the keystone activations with two types of representative **nonzero** values computed from that group:
> > > (1) the **average activation** over the generated response, and
> > > (2) the **activation at the first generated token**.
> > > If keystone neurons mainly acted as prompt-independent constants, then patching them with such realistic representative values should have preserved at least part of the model’s behavior. However, this is not what we observe: under **all** patching settings, performance collapses completely.
> > > |        Setting        | MMLU | Math500 | EvalPlus | MGSM |
> > > |:---------------------:|:----:|:-------:|:--------:|:----:|
> > > | Base Llama-3.1-8B-Instruct | 71.3 | 49.6 | 53.2 | 79.2 |
> > > | Using general\_task (average) | 0 | 0 | 0 | 0 |
> > > | Using math (average) | 0 | 0 | 0 | 0 |
> > > | Using code (average) | 0 | 0 | 0 | 0 |
> > > | Using multilingual (average) | 0 | 0 | 0 | 0 |
> > > | Using general\_task (first) | 0 | 0 | 0 | 0 |
> > > | Using math (first) | 0 | 0 | 0 | 0 |
> > > | Using code (first) | 0 | 0 | 0 | 0 |
> > > | Using multilingual (first) | 0 | 0 | 0 | 0 |
> > >
> > > This result suggests that keystone neurons are not merely providing a large constant offset. Instead, the model depends on their precise inference-time activations, indicating that they participate in the actual computation required for successful generation.
> > > We sincerely appreciate this thoughtful perspective and would be very glad to hear your view on this interpretation and continue the discussion.

---

### Official Review · Reviewer_cDhf · 2026-03-08

**Soundness:** 3
**Presentation:** 3
**Significance:** 3
**Originality:** 2
**Overall Recommendation:** 5
**Confidence:** 4

**Summary:**

The authors identify neurons that are activated the most by dufferent set of prompts. They compute this on a per-block basis. The show that a very few (typically <0.2%) are critical to the performance of multiple, diverse tasks, and removing them degrades performance. They call this set keystone neurons. The authors analyze these neurons, and they show that their variance is typically lower compared to the others, and the model is also more sensitive to perturbing these neurons compared to random neurons. Finally, the authors propose finetuning only these neurons and show that the model exhibits better performance on the target task and less catastrophic forgetting.

**Compliance With Llm Reviewing Policy:**

Affirmed.

**Final Justification:**

During the rebuttal, the authors performed activation patching experiments, which showed that the keystone neurons are not merely biased. The method provides an easy, cost-effective way to fine-tune a small subset of neurons while achieving superior performance compared to the baseline. While I feel that framing the findings as a "tiny brain" that does most of the processing is a strong statement and would require a paper on its own to analyze, I think the paper is already worth presenting to the community.

**Key Questions For Authors:**

- The terminology is sometimes unclear: what is "capability dimensions" or "module block"?
- Is the top-k taken on the neurons of the whole FFN/Attention block, or on each linear projection?
- Figure 1 is low contrast with small text and very hard to read.
- The authors forgot to change the running title on the internal pages.

**Limitations:**

yes

**Strengths And Weaknesses:**

Strengths:
- The technique proposed by the authors is simple and effective, and has direct practical use for better finetuning.
- The paper is mostly clearly written.

Weaknesses:
- Main weakness: In Tab 3, the math full finetuning makes the Llama-3.1-8B-Instruct worse on math than not doing it. How is this possible? Is the setup (learning rate, number of steps, batch size, etc.) correct? An alternative explanation for the findings of the paper is that the hyperparameters are wrong, and the masking makes the model less sensitive to wrong hyperparameters.
- More details on what exactly constitues as a "module-block" would be welcome. See questions.

The finding that the highly activated neurons are important is not very surprising alone, given that the outlier activations are well studied and understood as critical. However, it is surprising that fine-tuning these can lead to better performance and less catastrophic forgetting.

---

> ### Author Rebuttal · Authors · 2026-03-31
>
> **W1. Why does full math fine-tuning make Llama-3.1-8B-Instruct worse on math than not fine-tuning at all? Could this simply be due to wrong hyperparameters, with masking merely making the model less sensitive?**
>
> Thank you for raising this possibility. We agree that degraded math performance after math SFT can in principle be caused by poor hyperparameter choices, so we took care to tune learning rates and report the best-performing configuration for each method. More importantly, this phenomenon is not unique to our setup: recent work reports that fine-tuning instruction-tuned models can degrade reasoning or cross-domain performance[a,b], and attributes part of this effect to distribution shift and changes in internal reasoning behavior rather than simply failed optimization[a,c].  In our case, we used the same training recipe family as prior OpenMathInstruct2-style settings for Llama3.1-8B-Base[d], and we also observe normal-looking gains for Llama-3.2-1B-Instruct on math and for both models on the other downstream task, which makes a purely “wrong hyperparameters” explanation less likely. We therefore view hyperparameter sensitivity as a plausible contributing factor, but not a sufficient explanation of the overall pattern, and we will clarify this discussion in the revision.
>
> [a] Wu et al. Clear Minds Think Alike: What Makes LLM Fine-tuning Robust? A Study of Token Perplexity arXiv preprint arXiv:2501.14315v1, 2025.
>
> [b] Wu et al. On the Generalization of SFT: A Reinforcement Learning Perspective with Reward Rectification.  arXiv preprint arXiv:2508.05629, 2025
>
> [c]  Lobo et al. On the impact of fine-tuning on chain-of-thought reasoning. arXiv preprint arXiv:2411.15382, 2024
>
> [d] Toshniwal et al. OpenMathInstruct-2: Accelerating AI for Math with Massive Open-Source Instruction Data. arXiv preprint arXiv:2410.01560, 2024
>
> **Q1. What do “capability dimensions” and “module block” mean?**
>
> Thank you for pointing this out. By **capability dimensions**, we mean the broad ability categories used in our probing setup: general tasks, mathematical reasoning, code generation, and multilingual tasks, instantiated by **MMLU, MATH500, EvalPlus, and MGSM**, respectively. We agree that **module block** was imprecise, and we will replace it with **module** for clarity.
>
> **Q2. Is top-\(k\) taken over the whole FFN/attention block, or over each linear projection?**
>
> In our implementation, the top-\(k\) selection is performed within each individual **linear projection** (e.g., attention **Q/K/V/O** or FFN **up/down**), rather than over the entire attention or FFN component. We will clarify this explicitly in the paper.
>
> **Q3. Figure 1 is hard to read, and the running title on the internal pages was not updated.**
>
> Thank you for pointing this out. We will improve Figure 1 by increasing contrast and enlarging the text for better readability. We also thank the reviewer for noticing the running-title issue, and we will correct it in the revised version.

---

> > ### Author Rebuttal · Reviewer_cDhf · 2026-04-02
> >
> > Thank you for the rebuttal. Can the authors comment on how it is possible that tuning only the neurons with low activation variance can have such a large effect? Also, I think it needs to be rephrased, because it seems to be very unlikely that these are the neurons actually resposible for the majority of the computation.

---

> > > ### Author Response · Authors · 2026-04-04
> > >
> > > Thank you for this thoughtful comment. We would like to clarify that keystone neurons are not selected simply because they have **low activation variance**. Rather, they are identified as neurons that remain in a **consistently high-activation regime across diverse prompts and generated tokens**. In our view, this combination is important: their high activation suggests that they are actively involved throughout inference, while their cross-prompt stability indicates that this involvement is not limited to a narrow task-specific pattern.
> > >
> > > This also helps explain why tuning only this small subset can have a large effect. Because these neurons are already highly active across essentially all decoding steps, modifying them is more likely to influence the downstream task in a way that is actually used during generation. At the same time, restricting updates to this small but consistently engaged subset reduces the extent of global parameter drift, which may help preserve the model’s other capabilities better than full-parameter tuning.
> > >
> > > To further test whether high activation here really reflects meaningful participation in computation, rather than merely a large constant contribution, we performed an additional **activation patching** experiment on the keystone neurons of **Llama-3.1-8B-Instruct**. For each capability group (**general-task**, **math**, **code**, **multilingual**), we replaced the keystone activations with two types of representative **nonzero** values computed from that group:
> > > (1) the **average activation** over the generated response, and
> > > (2) the **activation at the first generated token**.
> > >
> > > If keystone neurons mainly acted as prompt-independent constants, then patching them with such realistic representative values should have preserved at least part of the model’s behavior. However, under **all** patching settings, performance collapses completely:
> > >
> > > | Setting | MMLU | Math500 | EvalPlus | MGSM |
> > > |:--|:--:|:--:|:--:|:--:|
> > > | Base Llama-3.1-8B-Instruct | 71.3 | 49.6 | 53.2 | 79.2 |
> > > | Using general_task (average) | 0 | 0 | 0 | 0 |
> > > | Using math (average) | 0 | 0 | 0 | 0 |
> > > | Using code (average) | 0 | 0 | 0 | 0 |
> > > | Using multilingual (average) | 0 | 0 | 0 | 0 |
> > > | Using general_task (first) | 0 | 0 | 0 | 0 |
> > > | Using math (first) | 0 | 0 | 0 | 0 |
> > > | Using code (first) | 0 | 0 | 0 | 0 |
> > > | Using multilingual (first) | 0 | 0 | 0 | 0 |
> > >
> > > This result suggests that keystone neurons are not merely providing a large constant offset. Instead, the model depends on their **precise inference-time activations**, which indicates that they participate in the actual computation required for successful generation.
> > >
> > > We also agree with the reviewer that the phrasing should be more careful. Our claim is **not** that keystone neurons are solely responsible for the majority of the model’s computation. Rather, we view them as a small but functionally critical subset whose participation is repeatedly required across diverse generations.

---

### Official Review · Reviewer_CwpX · 2026-03-08

**Soundness:** 3
**Presentation:** 3
**Significance:** 3
**Originality:** 3
**Overall Recommendation:** 4
**Confidence:** 4

**Summary:**

This paper shows that a small number of neurons can be identified in LLMs that seem to account for a large part of their performance. These neurons are identified using a procedure that tracks the absolute magnitude of activation across a relatively small set of prompts. Ablation of the resulting neurons substantially affects performance, and using models instantiating only a subset of neurons identified by the approach preserves much of the performance of the models.

**Compliance With Llm Reviewing Policy:**

Affirmed.

**Key Questions For Authors:**

1. Using |a| potentially selects for both neurons that vary a lot across prompts and neurons that are constantly high in activation across prompts. The authors note that the latter seem to be the more important ones. Is it possible to define a metric that better selects for those neurons, or more clearly separate out these two classes and provide analysis that supports this result?

2. In Figure 2, what is the accuracy without removing neurons? Accuracy on what?

3. At the top of page 4 we are told that "representative prompts" are used, how were these selected?

**Limitations:**

Limitations and societal impacts are appropriately discussed.

**Strengths And Weaknesses:**

Strengths:

This is an interesting analysis that identifies subsets of models that account for much of their performance with a relatively lightweight procedure. The analyses are generally convincing in showing that this subset is important. A good range of models is considered.

Weaknesses:

1. As the authors note, there is an existing literature showing that sparse subnetworks can capture a lot of the performance of neural networks, meaning that this result is not entirely unexpected. The method itself being relatively simple and working in large scale models is thus a key part of the contribution.

2. Understanding to what extent this is driven simply by the size of the weights (rather than their coherence) would be useful. This could be evaluated by running the same analyses on networks with randomly perturbed weights and assessing how the outputs correlate when the network is reduced to a subset of neurons as in the procedure outlined here. My guess is that the outputs will continue to correlate, suggesting that this isn't necessarily a consequence of training the models, but rather of how neural networks work.

3. The results are presented without error bars, confidence intervals, or statistical hypothesis tests. This makes it difficult to evaluate whether the results in Table 1 are genuinely different.

4. It's not clear how many random seeds are used in the fine-tuning analyses. Again, providing information about variation across seeds and error bars etc. for these results is important. (Applies to Tables 3 and 4)

---

> ### Author Rebuttal · Authors · 2026-03-31
>
> **W1. How is this different from prior sparse-subnetwork results?**
>
> Thank you for this comment. We agree that our findings are broadly consistent with prior work showing that sparse subsets of parameters can matter disproportionately. However, our setting and claim are different. We do **not** seek a recoverable sparse subnetwork that preserves performance under pruning. Instead, we identify a tiny set of concrete **neurons** whose deactivation causes **complete capability collapse**. Unlike prior work on task-specific neurons, our keystone neurons are selected by persistent engagement across multiple capability dimensions, suggesting a **cross-task structural neuron subset** rather than a mechanism tied to one domain. We will clarify this connection in the revision.
>
> **W2. Is the effect driven by weight magnitude rather than learned coherence?**
>
> Thank you for this suggestion. To test whether the effect is explained mainly by weight magnitude, we additionally ablated neurons selected purely by **parameter norm / weight magnitude**. This baseline is clearly weaker than keystone-neuron ablation: while norm-based ablation degrades performance, it does **not** cause complete capability collapse. For example, under norm-based ablation, Llama-3.1-8B-Instruct still retains **67.58 MMLU / 29.52 Math500 / 44.29 EvalPlus / 64.48 MGSM**, and Qwen2.5-7B-Instruct retains **45.19 / 10.80 / 20.00 / 32.40**, whereas keystone-neuron ablation drives all benchmark accuracies to **0**. This shows that the effect is **not** explained simply by large parameter norms, but reflects a more specific learned structural pattern. We will clarify this point in the revision and include the norm-based ablation control.
>
> **W3. Where are the error bars / confidence intervals / significance tests for Table 1?**
>
> Thank you for this suggestion. We agree that uncertainty reporting should be clearer. In our ablation experiments, we extract multiple keystone sets using different prompt groups. These sets have high overlap, and ablating them consistently drives benchmark accuracy to **0**, so the core conclusion of Table 1—**complete capability collapse after keystone ablation**—is highly stable. The main variation appears in **perplexity**, not in whether collapse occurs. We will clarify this protocol and report perplexity variability more explicitly in the revision.
>
> **W4. How many random seeds are used in the fine-tuning analyses (Tables 3 and 4)?**
>
> Thank you for this suggestion. We agree that seed variability is important. Our current Tables 3 and 4 do **not** report multi-seed averages. Instead, for each method we explored multiple learning rates and reported the **best-performing configuration**. We will clarify this protocol in the revision. Our goal is to compare the best achievable behavior of **full-parameter** vs. **keystone-only** fine-tuning under comparable tuning effort, rather than present a seed-variance study. We agree that multi-seed statistics would strengthen this part and will state this limitation more clearly.
>
> **Q1. Does using \|a\| mix together constantly high neurons and highly variable neurons? Can these be separated more clearly?**
>
> Thank you for this question. Our selection procedure already suppresses neurons that are highly active for only one prompt but unstable across prompts. Within each capability dimension, we rank neurons by activation, retain only the top-\(k\) set, and define the final candidate pool by the **intersection across prompts**. Therefore, a neuron must remain highly ranked across diverse prompts to be selected; large activation in only one prompt is not sufficient. In this sense, the procedure already biases toward neurons that are **consistently strongly engaged**. We will clarify this point and further emphasize our **mean–CV analysis**, which shows that the selected keystone neurons concentrate in a **high-activation, low-variability** regime.
>
> **Q2. In Figure 2, what is the accuracy without removing neurons, and accuracy on what?**
>
> Thank you for pointing this out. In Figure 2, the reported accuracy refers to the model’s **aggregate benchmark accuracy** over the four capability dimensions used in our study: **MMLU, MGSM, EvalPlus, and MATH500**. We will make this explicit in the figure caption and main text.
>
> **Q3. How were the “representative prompts” selected?**
>
> Thank you for pointing this out. Our representative prompts were chosen by sampling one instruction from each of the four benchmark families used to probe the corresponding capability dimensions: **MMLU** (general tasks), **MATH500** (mathematical reasoning), **EvalPlus** (coding), and **MGSM** (multilingual tasks). To verify that the results are not tied to a particular prompt choice, we repeated the detection procedure using **five disjoint prompt groups** from the same four benchmark families and observed high overlap among the resulting keystone sets. We will clarify this protocol in the revision.

---

> > ### Author Rebuttal · Reviewer_CwpX · 2026-04-01
> >
> > Thank you for the detailed rebuttal. This largely addresses my concerns and questions, but raises some others. Focusing on the best-performing configuration (and not including statistical hypothesis tests or error bars) raises the possibility that there is a high false-discovery rate -- it gives many opportunities to find a difference and magnifies the chance that a difference appears without a real underlying effect.

---

> > > ### Author Response · Authors · 2026-04-04
> > >
> > > Thank you for this important comment. We agree that reporting only the best-performing configuration can raise concerns about selection bias and an inflated false-discovery rate. We will clarify this point more carefully in the revision.
> > >
> > > For the **fine-tuning experiments**, our goal was not to search over a large hyper-parameter space until a favorable result appeared, but to compare **Full FT** and **Keystone-only FT** under the **same controlled optimization settings**. We varied only the learning rate over **{2e-5, 1e-5, 1e-6}**, while fixing the remaining hyper-parameters (**batch size = 32, weight decay = 1e-2, epochs = 2**). For each setting, we used a held-out **1,000-example in-distribution validation set** and selected the best checkpoint on that validation set for final evaluation. We will make this protocol explicit and include the full learning-rate table in the revised paper, rather than only highlighting the best configuration.
> > >
> > > More importantly,  **Keystone-only FT** is consistently much more stable than **Full FT**, while **Full FT** remains substantially below the Base model. Thus, the main comparison does not depend on a single favorable configuration.
> > >
> > > | Model | LR | Setting | Math500 | GSM8K | Olympiad | AIME-24 | MMLU | EvalPlus | MGSM |
> > > |:-----:|:--:|:-------:|:-------:|:-----:|:--------:|:-------:|:----:|:--------:|:----:|
> > > | Llama-3.1-8B-Instruct | – | **Base** | 49.60 | 83.40 | 16.59 | 3.33 | **71.3** | 53.2 | **77.2** |
> > > | -| 2e-5 | Full FT | 31.40 | 70.05 | 8.15 | 0.00 | 62.5 | 32.2 | 60.8 |
> > > | - | 2e-5 | **Keystone-only** | **51.20** | **84.31** | **16.74** | **6.67** | 69.8 | 55.0 | 76.6 |
> > > | - | 1e-5 | Full FT | 30.80 | 69.20 | 7.80 | 0.00 | 61.8 | 34.8 | 60.4 |
> > > | - | 1e-5 | **Keystone-only** | 50.55 | 83.60 | 16.20 | **6.67** | 70.1 | 56.6 | 76.0 |
> > > | - | 1e-6 | Full FT | 29.60 | 72.80 | 9.38 | 0.00 | 62.0 | 32.8 | 62.4 |
> > > |-| 1e-6 | **Keystone-only** | 48.00 | 84.20 | **16.74** | 3.33 | 70.92 | **55.8** | 76.8 |
> > >
> > > For the **keystone identification** itself, the result is also not based on one opportunistic search. In **Table 2**, we already report the overlap of keystone neurons obtained from **five disjoint prompt groups**, and the overlap is consistently very high. This indicates that the identified neurons are a **stable structural subset**, rather than an artifact of a particular prompt sample.
> > >
> > > In addition, for the ablation/replacement experiments, the qualitative effect is highly stable: across repeated runs, the downstream task accuracies are always **0**, while only the exact perplexity values vary. For example, under random replacement on **Llama-3.1-8B-Instruct**, all five seeds give **0** on MMLU, Math500, EvalPlus, and MGSM, while C4/WikiText-2 perplexities remain uniformly catastrophic despite some variation in magnitude:
> > >
> > > | Model / Setting | MMLU | Math500 | EvalPlus | MGSM | C4 | W2 |
> > > |:--|:--:|:--:|:--:|:--:|:--:|:--:|
> > > | Llama-3.1-8B-Instruct | 71.3 | 49.6 | 53.2 | 79.2 | 11.78 | 13.61 |
> > > | seed42 | 0 | 0 | 0 | 0 | 522.57 | 879.98 |
> > > | seed43 | 0 | 0 | 0 | 0 | 547.23 | 801.66 |
> > > | seed44 | 0 | 0 | 0 | 0 | 688.41 | 563.12 |
> > > | seed45 | 0 | 0 | 0 | 0 | 823.09 | 879.00 |
> > > | seed46 | 0 | 0 | 0 | 0 | 606.58 | 742.37 |
> > >
> > > So while we agree that statistical reporting should be improved, the central findings here are not fragile:
> > > (1) the **identified keystone sets are highly reproducible across prompt groups**, and
> > > (2) the **functional collapse after intervention is qualitatively invariant**, even when the exact perplexity values differ.
> > >
> > > We will revise the paper to make this robustness evidence more explicit, and we thank the reviewer for pointing out the need to present it more clearly.

---

### Official Review · Reviewer_gt2n · 2026-03-17

**Soundness:** 1
**Presentation:** 3
**Significance:** 2
**Originality:** 2
**Overall Recommendation:** 2
**Confidence:** 4

**Summary:**

The authors find that removing (by replacing their values with zero) a few neurons completely destroys model performance, in many tasks. These neurons can be identified from as little as 4 prompts.

**Compliance With Llm Reviewing Policy:**

Affirmed.

**Key Questions For Authors:**

- Is the "constant neurons" hypothesis I present actually false? Why does the (very small) variability of these "keystone neurons" matter? For example, you could copy the value of the neuron from a related but very different prompt and see the change in behavior (activation patching).
- Did you think of the hypothesis above, or is it new? What other hypotheses can you think of that explain your findings?

**Limitations:**

Yes

**Strengths And Weaknesses:**

**Soundness:** a good scientific paper considers all the likely hypotheses for the observed phenomenon, and conducts several experiments until only one reasonable hypothesis is left. Alternatively, the paper lists the many hypotheses that could explain the data, and acknowledges that experiments only discard some of them.

There's one very obvious hypothesis here in this paper that isn't examined at all. What if these "keystone neurons"... are simply *learned constants* that the model uses in the forward pass? Sections 2 and most of 3 don't consider this hypothesis, and even make it sound like they're selecting neurons that activate highly, but *differently* for each task: e.g. line 137-138L "we define a_i(p) as the average of |a_i(p, t)| over all the generated tokens", which implies that it can vary across tokens. Then "this quantity provides a simple measure of neuron i's contribution to the model's processing of prompt p; which implies that the neuron does some meaningful processing, because would constants do any processing?

Then in Section 3.4 we're hit with confirmation of this hypothesis, that basically is buried:  "keystone neurons lie in the tail region characterized by relatively large mean activation and a small coefficient of variation". **These neurons are therefore basically constant, how can they do any meaningful processing?**

The multiplicative rescaling sensitivity is also compatible with this "constant hypothesis" (changing the constants would make the model change behavior a lot, of course). Then Section 4 is interesting, it shows that altering these keystone neurons can change behavior of the LLM, perhaps because they behave like a steering vector that can be trained?

By the way, in the interpretability literature, we have long worried about whether a neuron is just a constant when attempting to attribute things to it. That's why [Zhang and Nanda (2023), a review paper,](https://arxiv.org/abs/2309.16042) dis-recommends using "zero ablation" to check the impact of neurons. (Replacing a neuron's value by zero, like this paper does, is zero ablation). Instead, the neurons' value from a related prompt is meant to be used. This makes it so that constant neurons like these basically do not show up in sensitivity analyses; which is good because they can't be doing any processing.

**Presentation:** good prose, thanks to AI. But some extraneous words, e.g. in line 212R: "keystone neurons concentrate on a narrow high-activation, low-variability frontier". What is a "frontier"? Does it just sound cool?.

I think the neurons should be called "constant neurons" if the above hypothesis is true.

**Significance:** low. Finetuning randomly scattered neurons is not computationally useful, and the presence of constants was known.

**Originality:** low.

---

> ### Author Rebuttal · Authors · 2026-03-31
>
> **Q1. Are keystone neurons merely learned constants, and is zero ablation sufficient?**
>
> Thank you for raising this important alternative hypothesis. We agree that highly activated neurons could, in principle, behave like learned constants, and that zero ablation alone does not fully rule this out.
>
> To directly test this possibility, we performed an additional **activation patching** experiment on the keystone neurons of **Llama-3.1-8B-Instruct**. For each capability group (general-task, math, code, multilingual), we replaced the keystone activations with two types of representative values computed from that group:
> (1) the **average activation over the generated response**, and
> (2) the **activation at the first generated token**.
>
> If keystone neurons mainly acted as prompt-independent constants, then patching them with such representative values should preserve at least part of the model’s behavior. However, this is not what we observe:
>
> |        Setting        | MMLU | Math500 | EvalPlus | MGSM |
> |:---------------------:|:----:|:-------:|:--------:|:----:|
> | Base Llama-3.1-8B-Instruct | 71.3 | 49.6 | 53.2 | 79.2 |
> | Using general\_task (average) | 0 | 0 | 0 | 0 |
> | Using math (average) | 0 | 0 | 0 | 0 |
> | Using code (average) | 0 | 0 | 0 | 0 |
> | Using multilingual (average) | 0 | 0 | 0 | 0 |
> | Using general\_task (first) | 0 | 0 | 0 | 0 |
> | Using math (first) | 0 | 0 | 0 | 0 |
> | Using code (first) | 0 | 0 | 0 | 0 |
> | Using multilingual (first) | 0 | 0 | 0 | 0 |
>
> Under **all** patching settings, performance collapses completely. If these neurons merely contributed a fixed constant offset, replacing them with realistic, nonzero representative values should have preserved at least partial functionality. Instead, the model depends on the **precise inference-time activations** of these neurons.
>
> To provide a control **distinct from zero ablation**, we further replaced each keystone neuron’s **parameters** with those of a random neuron from the **same layer and module**. This again causes complete collapse across benchmarks for both **Llama-3.1-8B-Instruct** and **Qwen2.5-7B-Instruct**:
>
> |                 Model / Setting                 | MMLU | Math500 | EvalPlus | MGSM | C4 | W2 |
> |:-----------------------------------------------:|:----:|:-------:|:--------:|:----:|:---:|:---:|
> | Llama-3.1-8B-Instruct | 71.3 | 49.6 | 53.2 | 79.2 | 11.78 | 13.61 |
> | Random replacement (avg over 5 seeds) | 0±0 | 0±0 | 0±0 | 0±0 | 3647.47±1452.48 | 4333.61±952.72 |
> | Qwen2.5-7B-Instruct | 75.7 | 74.8 | 63.2 | 87.0 | 12.42 | 12.56 |
> | Random replacement (avg over 5 seeds) | 0±0 | 0±0 | 0±0 | 0±0 | 12755.51±8526.60 | 6596.78±5899.67 |
>
> This complementary control shows that the effect is not specific to zero ablation: the **parameter configuration** of keystone neurons is itself crucial and cannot be replaced by random neurons from the same module.
>
> At the same time, the activation statistics show that keystone neurons occupy a **consistently high-activation regime** across diverse prompts, rather than behaving like sparse task-specific detectors:
>
> |      Keystone neuron      | General-task | Math | Code | Multilingual |
> |:-------------------------:|:------------:|:----:|:----:|:------------:|
> | fwd\_up (16,13860) | 4.169 | 3.62 | 3.52 | 3.60 |
> | fwd\_down (9,4255) | 0.49 | 0.59 | 0.65 | 0.59 |
> | atten\_q (18,2417) | 8.46 | 9.28 | 10.28 | 8.43 |
> | atten\_k (11,1134) | 10.54 | 12.11 | 12.33 | 11.27 |
> | atten\_v (18,2417) | 1.71 | 1.76 | 2.68 | 1.91 |
> | atten\_o (7,659) | 0.47 | 0.38 | 0.40 | 0.39 |
>
> These statistics suggest an important nuance: keystone neurons do remain in a stable high-activation regime across heterogeneous prompts, but they are **not** mere constants. If they were, patching with representative nonzero activations or replacing them with random same-module parameters should not have caused complete functional collapse.
>
> We therefore do **not** interpret keystone neurons as domain-specific “math/code neurons.” Rather, our current hypothesis is that they support more abstract, infrastructure-like computations repeatedly required across heterogeneous generations, such as maintaining coherent internal representations or stable generation dynamics.
>
> We will revise the paper to (1) explicitly acknowledge the limitation of zero ablation and this alternative hypothesis, and (2) add the activation patching, random-replacement, and activation-statistics evidence above to show that keystone neurons are not explained by a simple learned-constant account.

---

### Decision · Program_Chairs · 2026-04-30

**Decision:**

Accept (regular)

**Comment:**

Authors find that ablating a small number of "keystone" neurons can substantially degrade the performance of an LLM. They propose a lightweight method that relies on a few prompts to identify these neurons. They further propose a new fine-tuning approach that updates only the keystone neurons. All reviewers found the findings interesting.

During the rebuttal phase, authors brought up a few critical issues related to baselines, alternative interventions and controls.  Authors satisfactorily and convincingly responded to all these concerns with extra experiments.

Notably, reviewer gt2n proposed an alternative interpretation of the results in this paper, that the keystone neurons could have been learned constants, and hence they could not be involved in meaningful processing. Despite the reviewer not engaging in a discussion post-rebuttal, the AC is of the opinion that the rebuttal addressed this criticism.

Overall, the AC recommends acceptance.